# NADPEx: An on-policy temporally consistent exploration method for deep reinforcement learning

**Sirui Xie, Junning Huang, Chunxiao Liu, Lanxin Lei, Zheng Ma, Wei Zhang, Liang Lin**
SenseTime
`xiesirui@sensetime.com`
`{huangjunning, liuchunxiao, mazheng, wayne.zhang}@sensetime.com`
`linliang@ieee.org`

## Abstract

Reinforcement learning agents need exploratory behaviors to escape from local optima. These behaviors may include both immediate dithering perturbation and temporally consistent exploration. To achieve these, a stochastic policy model that is inherently consistent through a period of time is in desire, especially for tasks with either sparse rewards or long term information. In this work, we introduce a novel on-policy temporally consistent exploration strategy - Neural Adaptive Dropout Policy Exploration (NADPEx) - for deep reinforcement learning agents. Modeled as a global random variable for conditional distribution, dropout is incorporated to reinforcement learning policies, equipping them with inherent temporal consistency, even when the reward signals are sparse. Two factors, gradients' alignment with the objective and KL constraint in policy space, are discussed to guarantee NADPEx policy's stable improvement. Our experiments demonstrate that NADPEx solves tasks with sparse reward while naive exploration and parameter noise fail. It yields as well or even faster convergence in the standard mujoco benchmark for continuous control.

## 1 Introduction

Exploration remains a challenge in reinforcement learning, in spite of its recent successes in robotic manipulation and locomotion (Schulman et al., 2015b; Mnih et al., 2016; Duan et al., 2016; Schulman et al., 2017b). Most reinforcement learning algorithms explore with stochasticity in stepwise action space and suffer from low learning efficiency in environments with sparse rewards (Florensa et al., 2017) or long information chain (Osband et al., 2017). In these environments, temporally consistent exploration is needed to acquire useful learning signals. Though in off-policy methods, autocorrelated noises (Lillicrap et al., 2015) or separate samplers (Xu et al., 2018) could be designed for consistent exploration, on-policy exploration strategies tend to be learnt alongside with the optimal policy, which is non-trivial. A complementary objective term should be constructed because the purpose of exploration to optimize informational value of possible trajectories (Osband et al., 2016) is not contained directly in the reward of underlying Markov Decision Process (MDP). This complementary objective is known as reward shaping *e.g. optimistic towards uncertainty* heuristic and intrinsic rewards. However, most of them require complex additional structures and strong human priors when state and action spaces are intractable, and introduce unadjustable bias in *end-to-end* learning (Bellemare et al., 2016; Ostrovski et al., 2017; Tang et al., 2017; Fu et al., 2017; Houthooft et al., 2016; Pathak et al., 2017).

It would not be necessary though, to teach agents to explore consistently through reward shaping, if the policy model inherits it as a prior. This inspires ones to disentangle policy stochasticity, with a structure of time scales. One possible solution is policy parameter perturbation in a large time scale. Though previous attempts were restricted to linear function approximators (Rückstieß et al., 2008; Osband et al., 2014), progress has been made with neural networks, through either network section duplication (Osband et al., 2016) or adaptive-scale parameter noise injection (Plappert et al., 2017; Fortunato et al., 2017). However, in Osband et al. (2016) the episode-wise stochasticity is

unadjustable, and the duplicated modules do not cooperate with each other. Directly optimizing the mean of distribution of all parameters, Plappert et al. (2017) adjusts the stochasticity to the learning progress heristically. Besides, as all sampled parameters need to be stored in on-policy exploration, it is not directly salable to large neural networks, where the trade-off between large memory for multiple networks and high variance with sinlge network is non-trivial to make.

This work proposes *Neural Adaptive Dropout Policy Exploration* (NADPEx) as a simple, scalable and improvement-guaranteed method for on-policy temporally consistent exploration, which generalizes Osband et al. (2016) and Plappert et al. (2017). Policy stochasticity is disentangled into two time scales, step-wise action noise and episode-wise stochasticity modeled with a distribution of subnetworks. With one single subnetwork in an episode, it achieves inherent temporal consistency with only a little computational overhead and no crafted reward, even in environments with sparse rewards. And with a set of subnetworks in one sampling batch, agents experience diverse behavioral patterns. This sampling of subnetworks is executed through dropout (Hinton et al., 2012; Srivastava et al., 2014), which encourages composability of constituents and facilitates the emergence of diverse maneuvers. As dropout could be excerted on connections, neurons, modules and paths, NADPEx naturally extends to neural networks with various sizes, exploiting modularity at various levels. To align separately parametrized stochasticity to each other, this sampling is made differentiable for all possible dropout candidates. We further discuss the effect of a first-order KL regularizer on *dropout policies* to improve stability and guarantee policy improvement. Our experiments demonstrate that NAPDEx solves challenging exploration tasks with sparse rewards while achieving as efficient or even faster convergence in the standard mujoco benchmark for state-of-the-art PPO agents, which we believe can be generalized to any deep reinforcement learning agents.

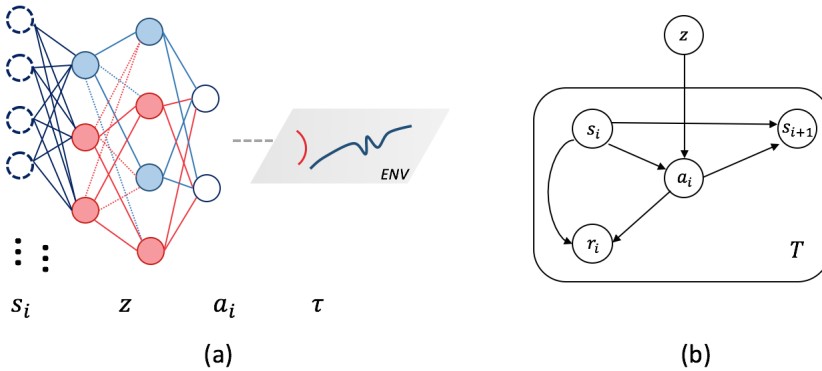

(a)    (b)

Figure 1: (a) Conceptual graph to illustrate the hierarchical stochasticity in NADPEx. Agent spontaneously explores towards different directions with different dropouts. (b) Graphical model for an MDP with NADPEx. $z$ is a global random variable in one episode $\tau$.

## 2 PRELIMINARIES

We consider a standard discrete-time finite-horizon discounted Markov Decision Process (MDP) setting for reinforcement learning, represented by a tuple $\mathcal{M} = (\mathcal{S}, \mathcal{A}, \mathcal{P}, r, \rho_0, \gamma, T)$, with a state set $\mathcal{S}$, an action set $\mathcal{A}$, a transitional probability distribution $\mathcal{P} : \mathcal{S} \times \mathcal{A} \times \mathcal{S} \to \mathbb{R}_+$, a bounded reward function $r : \mathcal{S} \times \mathcal{A} \to \mathbb{R}_+$, an initial state distribution $\rho_0 : \mathcal{S} \to \mathbb{R}_+$, a discount factor $\gamma \in [0, 1]$ and horizon $T$. We denote a policy parametrized by $\boldsymbol{\theta}$ as $\pi_{\boldsymbol{\theta}} : \mathcal{S} \times \mathcal{A} \to \mathbb{R}_+$. The optimization objective of this policy is to maximize the expected discounted return $\eta(\pi_{\boldsymbol{\theta}}) = \mathbb{E}_\tau[\sum_{t=0}^{T} \gamma^t r(s_t, a_t)]$, where $\tau = (s_0, a_0, ...)$ denotes the whole trajectory, $s_0 \sim \rho_0(s_0)$, $a_t \sim \pi_{\boldsymbol{\theta}}(a_t|s_t)$ and $s_{t+1} \sim \mathcal{P}(s_{t+1}|s_t, a_t)$. In experimental evaluation, we follow normal setting and use undiscounted return $\mathbb{E}_\tau[\sum_{t=0}^{T} r(s_t, a_t)]$.

### 2.1 PROXIMAL POLICY OPTIMIZATION (PPO)

Theoretically speaking, NADPEx works with any policy gradient based algorithms. In this work, we consider the recent advance in on-policy policy gradient method, proximal policy optimization (PPO) (Schulman et al., 2017b).

Policy gradient methods were first proposed by Williams (1992) in REINFORCE algorithm to maximize the aforementioned objective in a gradient ascent manner. The gradient of an expectation $\nabla_{\boldsymbol{\theta}}\eta(\pi_{\boldsymbol{\theta}})$ is approximated with a Monte Carlo average of policy gradient:

$$\nabla_{\boldsymbol{\theta}}\eta(\pi_{\boldsymbol{\theta}}) \approx \frac{1}{NT}\sum_{i=1}^{N}\sum_{t=o}^{T}\nabla_{\boldsymbol{\theta}}\log\pi_{\boldsymbol{\theta}}(a_t^i|s_t^i)(R_t^i - b_t^i), \tag{1}$$

where $N$ is the number of episodes in this batch, $R_t^i = \sum_{t'=t}^{T}\gamma^{t'-t}r_{t'}^i$ and $b_t^i$ is a baseline for variance reduction.

Schulman et al. (2015b) proposes a policy iteration algorithm and proves its monotonicity in improvement. With the $\pi_{\boldsymbol{\theta}}$'s corresponding discounted state-visitation frequency $\rho_{\boldsymbol{\theta}}$, the sampling policy $\pi_{\boldsymbol{\theta}^{old}}$ and the updated policy $\pi_{\boldsymbol{\theta}}$, it solves the following constrained optimization problem with a line search for an appropriate step size within certain bound $\delta_{KL}$, called *trust region*:

$$argmax_{\boldsymbol{\theta}} \quad \mathbb{E}_{s\sim\rho_{\boldsymbol{\theta}^{old}},a\sim\pi_{\boldsymbol{\theta}^{old}}}\Big[\frac{\pi_{\boldsymbol{\theta}}(a|s)}{\pi_{\boldsymbol{\theta}^{old}}(a|s)}A_{\boldsymbol{\theta}^{old}}(s,a)\Big]$$
$$s.t. \quad \mathbb{E}_{s\sim\rho_{\boldsymbol{\theta}^{old}}}[D_{KL}(\pi_{\boldsymbol{\theta}}^{old}(\cdot|s)|\pi_{\boldsymbol{\theta}}(\cdot|s))] \leq \delta_{KL} \tag{2}$$

where $D_{KL}(\cdot||\cdot)$ is the Kullback-Leibler (KL) divergence, $A_{\boldsymbol{\theta}^{old}}(s,a)$ is calculated with Generalized Advantage Estimation (GAE) (Schulman et al., 2015c).

Proximal Policy Optimization (PPO) transforms (2) to a unconstrained optimization and solves it with first-order derivatives, embracing established stochastic gradient-based optimizer like Adam (Kingma & Ba, 2014). Noting that $D_{KL}(\pi_{\boldsymbol{\theta}}^{old}(\cdot|s)|\pi_{\boldsymbol{\theta}}(\cdot|s))$ is actually a relaxation of *total variational divergence* according to Schulman et al. (2015b), whose first-order derivative is 0 when $\boldsymbol{\theta}$ is close to $\boldsymbol{\theta}^{old}$, $D_{KL}(\pi_{\boldsymbol{\theta}}(\cdot|s)||\pi_{\boldsymbol{\theta}}^{old}(\cdot|s))$ is a natural replacement. Combining the first-order derivative of $D_{KL}(\pi_{\boldsymbol{\theta}}(\cdot|s)||\pi_{\boldsymbol{\theta}}^{old}(\cdot|s))$ (Schulman et al., 2017a) (check Appendix A for a proof):

$$\nabla_{\boldsymbol{\theta}}D_{KL}(\pi_{\boldsymbol{\theta}}(\cdot|s)|\pi_{\boldsymbol{\theta}^{old}}(\cdot|s))] = \mathbb{E}_{a\sim\pi_{\boldsymbol{\theta}}}[\nabla_{\boldsymbol{\theta}}\log\pi_{\boldsymbol{\theta}}(a|s)(\log\pi_{\boldsymbol{\theta}}(a|s) - \log\pi_{\boldsymbol{\theta}^{old}}(a|s))], \tag{3}$$

PPO optimizes the following unconstrained objective, called KL PPO loss[1]:

$$\mathcal{L}_{KL} = \mathbb{E}_{s\sim\rho_{\boldsymbol{\theta}^{old}},a\sim\pi_{\boldsymbol{\theta}^{old}}}[r_{\boldsymbol{\theta}}(s,a)A_{\boldsymbol{\theta}^{old}}(s,a) - \frac{\beta}{2}(\log\pi_{\boldsymbol{\theta}}(a|s) - \log\pi_{\boldsymbol{\theta}^{old}}(a|s))^2], \tag{4}$$

where $r_{\boldsymbol{\theta}}(s,a) = \frac{\pi_{\boldsymbol{\theta}}(a|s)}{\pi_{\boldsymbol{\theta}^{old}}(a|s)}$. Schulman et al. (2017b) also proposes a clipping version PPO, as a lower bound to (4):

$$\mathcal{L}_{clip} = \mathbb{E}_{s\sim\rho_{\boldsymbol{\theta}^{old}},a\sim\pi_{\boldsymbol{\theta}^{old}}}[min(r_{\boldsymbol{\theta}}(s,a), clip(r_{\boldsymbol{\theta}}(s,a), 1-\epsilon, 1+\epsilon))A_{\boldsymbol{\theta}^{old}}(s,a)] \tag{5}$$

In this work, we try both KL PPO and clipping PPO.

## 2.2 DROPOUT

Dropout is a technique used in deep learning to prevent features from co-adaptation and parameters from overfitting, by randomly dropping some hidden neuron units in each round of feed-forward and back-propagation (Hinton et al., 2012; Srivastava et al., 2014). This is modeled by multiplying a Bernoulli random variable $z_j^k$ to each hidden unit, *i.e.* neuron activation $h_j^k$, for $j = 1...m$, where $m$ is the number of hidden units in $k$th layer. Then the neuron activation of the $k + 1$th layer is

$$\mathbf{h}^{k+1} = \sigma(\mathbf{W}^{(k+1)T}\mathbf{D}_z^k\mathbf{h}^k + \mathbf{b}^{(k+1)}), \tag{6}$$

where $\mathbf{D}_z = diag(\mathbf{z}) \in \mathbb{R}^{m\times m}$, $\mathbf{W}^{(k+1)}$ and $\mathbf{b}^{(k+1)}$ are weights and biases at $k + 1$th layer respectively and we simply denote them with $\boldsymbol{\theta} \doteq <\mathbf{W}, \mathbf{b}>$. $\sigma(x)$ is the nonlinear neuron activation function. The parameter of this Bernoulli random variable is $p_j^k \doteq \mathcal{P}(z_j^k = 0)$, *aka the dropout rate* of the $j$th hidden unit at $k$th layer. In supervised learning, these dropout parameters are normally fixed during training in some successful practice (Hinton et al., 2012; Wan et al., 2013). And there are some variants to dropout connections, modules or paths (Wan et al., 2013).

---

[1]Though the concrete form is not provided in Schulman et al. (2017b), it is given in Dhariwal et al. (2017) publicly released by the authors

## 3 METHODOLOGY

### 3.1 NEURAL ADAPTIVE DROPOUT POLICY EXPLORATION (NADPEx)

Designing an exploration strategy is to introduce a kind of stochasticity during reinforcement learning agents' interaction with the environment to help them get rid of some local optima. While action space noise (parametrized as a stochastic policy $\pi_{\theta}$) might be sufficient in environments where step-wise rewards are provided, they have limited effectiveness in more complex environments (Florensa et al., 2017; Plappert et al., 2017). As their complement, an exploration strategy would be especially beneficial if it can help either sparse reward signals (Florensa et al., 2017) or significant long-term information (Osband et al., 2016) to be acquired and back-propagated through time (BPTT) (Sutton & Barto, 1998). This motivates us to introduce a hierarchy of stochasticity and capture temporal consistency with a separate parametrization.

Our algorithm, *Neural Adaptive Dropout Policy Exploration* (NADPEx), models stochasticity at large time scales with a distribution of plausible subnetworks. For each episode, one specific subnetworks is drawn from this distribution. Inspected from a large time scale, this encourages exploration towards different directions among different episodes in one batch of sampling. And from a small time scale, temporal correlation is enforced for consistent exploration. Policies with a hierarchy of stochasticity is believed to represent a complex action distribution and larger-scale behavioral patterns (Florensa et al., 2017).

We achieve the drawing of subnetwork through dropout. As introduced in Section 2.2, originally, dropout is modeled through multiplying a binary random variable $z_j^k \sim Bernoulli(1 - p_j^k)$ to each neuron activation $h_j^k$. In Srivastava et al. (2014), this *binary dropout* is softened as continuous random variable dropout, modeled with a Gaussian random variable $z \sim \mathcal{N}(\mathbf{I}, \sigma^2)$, normally referred to as *Gaussian multiplicative dropout*. Here we denote both distributions with $q_{\phi}(z)$. Later we will introduce how both of them could be made differentibale and thus adaptive to learning progress.

During the sampling process, at the beginning of every episode $\tau^i$, a vector of dropout random variables $z^i$ is sampled from $q_{\phi}(z)$, giving us a *dropout policy* $\pi_{\theta|z^i}$ for step-wise action distribution. $z^i$ is fed to the stochastic computation graph (Schulman et al., 2015a) for the whole episode until termination, when a new round of drawing initiates. Similar as observations, actions and rewards, this random variable is stored as sampled data, which will be fed back to the stochastic computation graph during training. Policies with this hierarchy of stochasticity, *i.e.* NADPEx policies, can be represented as a joint distribution:

$$\pi_{\theta,\phi}(\cdot, z) = q_{\phi}(\mathbf{z})\pi_{\theta|z}(\cdot),\tag{7}$$

making the the objective:

$$\eta(\pi_{\theta,\phi}) = \mathbb{E}_{\mathbf{z} \sim q(\mathbf{z})}[\mathbb{E}_{\tau|\mathbf{z}}[\sum_{t=0}^{T} \gamma^t r(s_t, a_t)]].\tag{8}$$

If we only use the stochasticity of network architecture as a bootstrap, as in BootstrapDQN (Osband et al., 2016), the bootstrapped policy gradient training is to update the network parameters $\theta$ with the following gradients:

$$\nabla_{\theta}\eta(\pi_{\theta,\phi}) = \nabla_{\theta}\mathbb{E}_{\mathbf{z} \sim q_{\phi}(\mathbf{z})}[\eta(\pi_{\theta|z})] = \mathbb{E}_{\mathbf{z} \sim q_{\phi}(\mathbf{z})}[\nabla_{\theta}\eta(\pi_{\theta|z})] \approx \frac{1}{N}\sum_{i=1}^{N}\nabla_{\theta}\mathcal{L}(\theta, z^i)\tag{9}$$

$\mathcal{L}(\theta, z^i)$ is the surrogate loss of *dropout policy* $\pi_{\theta,z^i}$, variates according to the specific type of reinforcement learning algorithm. In next subsection, we discuss some pitfalls of bootstrap and provide and our solution to it.

### 3.2 TRAINING NADPEx

STOCHASTICITY ALIGNMENT

One concern in separating the parametrization of stochasticity in different levels is whether they can adapt to each other elegantly to guarantee policy improvement in terms of objective (8). Policy

gradients in (9) alway reduces $\pi_{\boldsymbol{\theta}|\boldsymbol{z}}$'s entropy (i.e. stochasticity at small time scale) as the policy improves, which corresponds to the increase in agent's certainty. But this information is not propagated to $q_{\boldsymbol{\phi}}$ for stochasticity at large time scale in bootstrap. As in this work $q_{\boldsymbol{\phi}}$ is a distribution of subnetworks, this concern could also be intuitively understood as component space composition may not guarantee the performance improvement in the resulting policy space, due to the complex non-linearity of reward function. Gangwani & Peng (2017) observe that a naive crossover in the parameter space is expected to yield a low-performance composition, for which an extra policy distillation stage is designed.

We investigate likelihood ratio trick (Ranganath et al., 2014; Schulman et al., 2015a) for gradient back-propagation from the same objective (8) through discrete distribution *e.g. binary dropout* and reparametrization trick (Kingma & Welling, 2013; Rezende et al., 2014) for continuous distribution *e.g. Gaussian multiplicative dropout*, thus covering all possible dropout candidates (The proof is provided in Appendix B):

$$\nabla_{\boldsymbol{\theta},\boldsymbol{\phi}}\eta(\pi_{\boldsymbol{\theta},\boldsymbol{\phi}}) \approx \frac{1}{N}\sum_{i=1}^{N}(\nabla_{\boldsymbol{\theta}}\mathcal{L}(\boldsymbol{\theta},\boldsymbol{z}^i) + \nabla_{\boldsymbol{\phi}}\log q_{\boldsymbol{\phi}}(\boldsymbol{z}^i)A(s_0^i,a_0^i)), \tag{10}$$

$$\nabla_{\boldsymbol{\theta},\boldsymbol{\phi}}\eta(\pi_{\boldsymbol{\theta},\boldsymbol{\phi}}) \approx \frac{1}{N}\sum_{i=1}^{N}\nabla_{\boldsymbol{\theta},\boldsymbol{\phi}}\mathcal{L}(\boldsymbol{\theta},\mathbf{I}+\boldsymbol{\phi}\odot\boldsymbol{\epsilon}^i). \tag{11}$$

In (10) $\boldsymbol{\phi}$ is the parameters for Bernoulli distributions, $A(s_0^i,a_0^i)$ is the GAE (Schulman et al., 2015c) for the $i$th trajectory from the beginning, *i.e.* $(s_0^i, a_0^i)$. In (11) $\boldsymbol{\phi} = \boldsymbol{\sigma}$ thus $\boldsymbol{z} = \mathbf{I} + \boldsymbol{\phi}\odot\boldsymbol{\epsilon}^i$, $\odot$ is an element-wise multiplication, and $\boldsymbol{\epsilon}^i$ is the dummy random variable $\boldsymbol{\epsilon}^i \sim \mathcal{N}(\mathbf{0},\mathbf{I})$.

POLICY SPACE CONSTRAINT

However, simply making the dropout ditribution differentiable may not guarantee the policy improment. As introduced in Section 2.1, Schulman et al. (2015b) reduce the monotonic policy improvement theory to a contraint on KL divergence in policy space. In this work, the analytical form of NADPEx policy $\pi_{\boldsymbol{\theta},\boldsymbol{\phi}}$ is obtainable as $q_{\boldsymbol{\phi}}$ is designed to be fully factorizable. Thus ones can leverage the full-fledged KL constraint to stablize the training of NADPEx.

Here we give an example of PPO-NADPEx. Similar as (3), we leverage a first-order approximated KL divergence for NADPEx policies. As credit assignment with likelihood ratio trick in (3) may suffer from *curse of dimensionality* for $q_{\boldsymbol{\phi}}(\boldsymbol{z})$, we stop gradients *w.r.t.* $\boldsymbol{\phi}$ from the KL divergence to reduce variance with a little bias. We further replace $\pi_{\boldsymbol{\theta},\boldsymbol{\phi}}$ with a representative, the *mean policy* $\pi_{\boldsymbol{\theta}} = \pi_{\boldsymbol{\theta}|\overline{\boldsymbol{z}}}$, where $\overline{\boldsymbol{z}}$ is the mean of dropout random variable vectors. Then the objective is:

$$\mathcal{L}_{KL}^{NADPEx} = \mathbb{E}[r_{\boldsymbol{\theta}|\boldsymbol{z}}(s,a)A_{\boldsymbol{\theta}^{old}}(s,a) - \frac{\beta}{2}(\log\pi_{\boldsymbol{\theta}}(a|s) - \log\pi_{\boldsymbol{\theta}^{old}|\boldsymbol{z}}(a|s))^2] \tag{12}$$

where $r_{\boldsymbol{\theta}|\boldsymbol{z}}(s,a) = \frac{\pi_{\boldsymbol{\theta}|\boldsymbol{z}}(a|s)}{\pi_{\boldsymbol{\theta}^{old}|\boldsymbol{z}}(a|s)}$. A proof for the second term is in Appendix C. Intuitively, KL divergence between *dropout polices* $\pi_{\boldsymbol{\theta}|\boldsymbol{z}}$ and *mean policies* $\pi_{\boldsymbol{\theta}}$ is added to remedy the omission of $\nabla_{\boldsymbol{\phi}}D_{KL}$. Optimizing the lower bound of (12), clipping PPO could adjust the clip ratio accordingly.

### 3.3 RELATIONS TO PARAMETER NOISE

Episode-wise parameter noise injection (Plappert et al., 2017) is another way to introduce the hierarchy of stochasticity, which could be regarded as a special case of NADPEx, with *Gaussian mulitplicative dropout* on connection and a heurstic adaptation of variance. That dropout at neurons is a local reparametrization of noisy networks is proved in Kingma et al. (2015). A replication of the proof is provided in Appendix D. They also prove this local reparametrization trick has a lower variance in gradients. And this reduction in variance could be enhanced if ones choose to drop modules or paths in NADPEx.

More importantly, as we scope our discussion down to on-policy exploration strategy, where $\boldsymbol{z}$ from $q_{\boldsymbol{\phi}}(\boldsymbol{z})$ need to be stored for training, NADPEx is more scalable to much larger neural networks with possibly larger mini-batch size for stochastic gradient-based optimizers. As a base case, to dropout neurons rather than connections, NADPEx has a complexity of $O(Nm)$ for both sampling and memory, comparing to $O(Nm^2)$ of parameter noise, with $m$ denoting the number of neurons in one layer. This reduction could also be enhanced if ones choose to drop modules or paths.

## 4 EXPERIMENTS

In this section we evaluate NADPEx by answering the following questions through experiments.

(i) Can NADPEx achieve state-of-the-art result in standard benchmarks?

(ii) Does NADPEx drive temporally consistent exploration in sparse reward environments?

(iii) Is the *end-to-end* training of the hierarchical stochasticity effective?

(iv) Is KL regularization effective in preventing divergence between *dropout policies* and the *mean policy*?

We developed NADPEx based on `openai/baselines` (Dhariwal et al., 2017). We run all experiments with PPO as reinforcement learning algorithm. Especially, we developed NADAPEx based on the GPU optimized version PPO. Details about network architecture and other hyper-parameters are available in Appendix E. During our implemented parameter noise in the same framework as NADPEx, we encountered *out of memory* error for a mini-batch size 64 and training batch size 2048. This proves our claim of sampling/memory complexity reduction with NADPEx. We had to make some trade-off between GPU parallelism and more iterations to survive.

For all experiment with NADPEx, we test the following configurations for the dropout hyper-parameter, the initial *dropout rate*: (a) $p_j^k = 0.01$, (b) $p_j^k = 0.1$, (c) $p_j^k = 0.3$. For *Gaussiasn dropout*, we estimate the *dropout rate* as $p_j^k = \sigma_j^k/(1 + \sigma_j^k)$. And for experiments with parameter noise, we set the initial variance accordingly: (a) $\sigma_j^k = 0.001$, (b) $\sigma_j^k = 0.01$, (c) $\sigma_j^k = 0.05$, to ensure same level of stochasticity. For experiments with fixed KL version PPO, we run with $\beta = 0.0005$. All figures below show statistics of experiments with 5 randomly generated seeds.

### 4.1 NADPEx IN STANDARD ENVIRONMENTS

First, we evaluate NADPEx's overall performance in standard environments on the continuous control environments implemented in OpenAI Gym (Brockman et al., 2016). From Figure 2 we can see that they show similar pattern with *Gaussian dropout* and *binary dropout*, given identical *dropout rates*. Between them, *Gaussian dropout* is slightly better at scores and consistency among initial *dropout rates*. With the three initial *dropout rates* listed above, we find $p_j^k = 0.1$ shows constistent advantage over others. On the one hand, when the initial *dropout rate* is small ($p_j^k = 0.01$), NADPEx agents learn to earn reward faster than agents with only action space noise. It is even possible that these agents learn faster than ones with $p_j^k = 0.1$ in the beginning. However, their higher variance between random seeds indicates that some of them are not exploring as efficiently and the NADPEx policies may not be optimal, therefore normally they will be surpassed by ones with $p_i^k = 0.1$ in the middle stage of training. On the other hand, large initial *dropout rate* ($p_j^k = 0.3$) tends to converge slowly, possibly due to the claim in Molchanov et al. (2017) that a large dropout rate could induce large varaince in gradients, making a stable convergence more difficult.

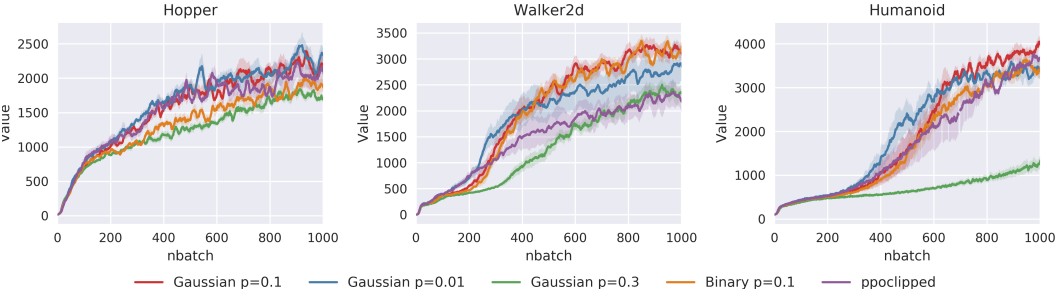

Figure 2: Experiments with NADPEx in standard envs, three $p_j^k$ are presented for *Gaussian dropout*, as well as the best of *binary dropout*. Extensive comparison is given in Appendix F.

## 4.2 TEMPORALLY CONSISTENT EXPLORATION

We then evaluate how NADPEx with *Gaussian dropout* performs in environments where reward signals are sparse. Comparing with environments with stepwise rewards, these environments are more challenging as they only yield non-zero reward signals after milestone states are reached. Temporally consistent exploration therefore plays a crucial role in these environments. As in Plappert et al. (2017) we run experiments in `rllab` (Duan et al., 2016), modified according to Houthooft et al. (2016). Specifically, we have: (a) *SparseDoublePendulum*, which only yields a reward if the agent reaches the upright position, and (b) *SparseHalfCheetah*, which only yields a reward if the agent crosses a target distance, (c) *SparseMountainCar*, which only yields a reward if the agent drives up the hill. We use the same time horizon of T = 500 steps before resetting and gradually increment the difficulty during repetition until the performance of NADPEx and parameter noise differentiates. We would like to refer readers to Appendix G for more details about the sparse reward environments.

As shown in Figure 3, we witness success with temporally consistent exploration through NAD-PEx, while action perturbation fails. In all enviroments we examed, larger initial *dropout rates* can achieve faster or better convergence, revealing a stronger exploration capability at large time scales. Comparing to parameter noise, NADPEx earns higher score in shorter time, possibly indicating a higher exploration efficiency.

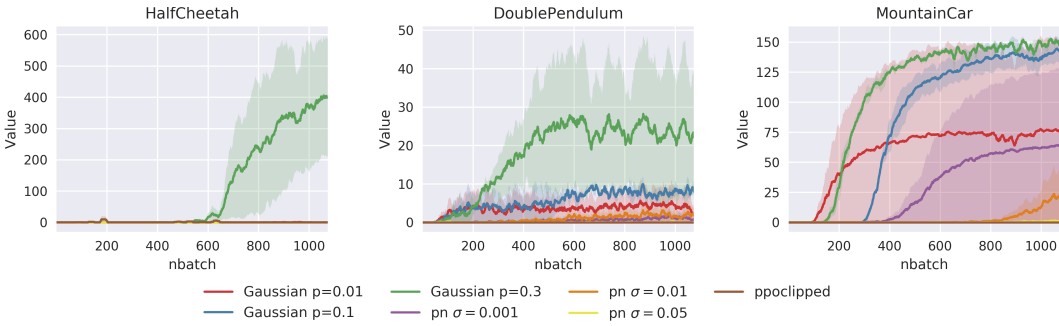

Figure 3: Experiments with NADPEx and parameter noise in sparse reward envs.

## 4.3 EFFECTIVENESS OF STOCHASTICITY ADAPTATION

One hypothesis NADPEx builds upon is *end-to-end* training of separately parametrized stochasticity can appropriately adapt the temporally-extended exploration with the current learning progress. In Figure 4 we show our comparison between NADPEx and bootstrap, as introduced in Section 3. And we can see that though the difference is subtle in simple task like *Hopper*, the advantage NADPEx has over bootstrap is obvious in *Walker2d*, which is a task with higher complexity. In *Humanoid*, the task with highest dimension of actions, the empirical result is interesting. Though bootstrap policy learns almost as fast as NADPEx in the begining, that the dropout is not adaptative drives it to over-explore when some *dropout policies* are close to converge. Being trapped at a local optimum, bootstrap policy earns 500 scores less than NADPEx policy.

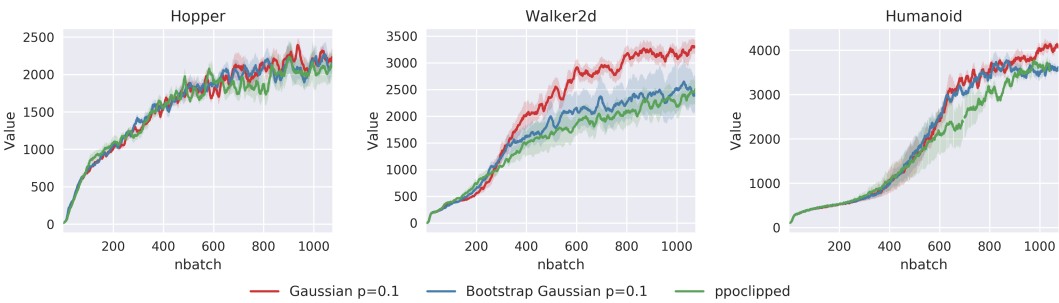

Figure 4: Comparison between NADPEx and bootstrap with *Gaussian dropout*.

### 4.4 KL DIVERGENCE BETWEEN DROPOUT POLICIES

To evaluate the effect of the KL regularizer, we also run experiments with KL PPO. Though in the original paper of Schulman et al. (2017b) clipping PPO empirically performs better than KL PPO, we believe including this KL term explicitly in the objective makes our validation self-explanatory. Figure 5 left shows the experiment result in *Walker2d*. Different from clipping PPO, NADPEx with small initial *dropout rate* performs best, earning much higher score than action noise. As shown in Figure 5 right, the KL divergence between *dropout polices* and *mean policies* is bounded.

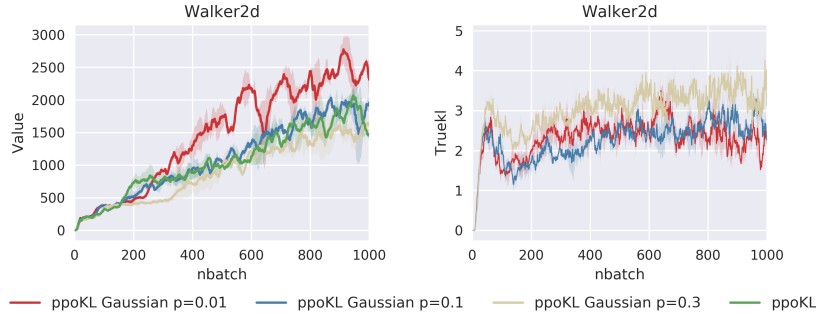

Figure 5: NADPEx KL PPO in *Walker2d*. Left: learning curves; right: true $D_{KL}(\pi_{\boldsymbol{\theta}^{old}|\boldsymbol{z}}||\pi_{\boldsymbol{\theta}|\overline{\boldsymbol{z}}})$.

## 5 RELATED WORKS

On-policy reinforcement learning methods have gained attention in recent years (Schulman et al., 2015b; Mnih et al., 2016; Schulman et al., 2017b), mainly due to their elegance with theoretical grounding and stability in policy iteration (Henderson et al., 2017b). Despite of the effectiveness, to improve their data efficiency remains an active research topic.

In this work, we consider the exploration strategies for on-policy reinforcement learning methods. Most of the aforementioned works employ naive exploration, with stochasticity only in action space. However, they fail to tackle some tasks with either sparse rewards (Florensa et al., 2017) or longterm information (Osband et al., 2016), where temporally consistent exploration is needed.

One solution to this challenge is to shape the reward to encourage more directed exploration. The specific *direction* has various foundations, including but not restricted to state visitation count (Jaksch et al., 2010; Tang et al., 2017), state density (Bellemare et al., 2016; Ostrovski et al., 2017; Fu et al., 2017), self-supervised prediction error (Pathak et al., 2017) etc. Some of them share the Probably Approximately Correct (PAC) with discrete and tractable state space (Jaksch et al., 2010). But when state space and action space are intractable, all of them need additional computational structures, which take non-trivial efforts to implement and non-negligible resources to execute.

Orthogonal to them, methods involving a hierarchy of stochasticity are proposed. Based on hierarchical reinforcement learning, Florensa et al. (2017) models the stochasticity at large time scales with a random variable - *option* - and model low-level policies with Stochastic Neural Networks. However, authors employ human designed proxy reward and staged training. Almost concurrently, Osband et al. (2016) and Plappert et al. (2017); Fortunato et al. (2017) propose network section ensemble and parameter noise injection respetively to disentangle stochasticity. Under the banner of Stochastic Neural Networks, NADPEx generalize them all. BootstrapDQN is a special case of NADPEx without stochasticity adapation and parameter noise is a special case with high variance and complexlity, as well as some heursitic approximation. Details are discussed in Section 3.

In NADPEx, this stochasticity at large time scales is captured with a distribution of plausible neural subnetworks from the same complete network. We achieve this through dropout (Srivastava et al., 2014). In spite of its success in supervised deep learning literature and Bayesian deep learning literature, it is the first time to combine dropout to reinforcement learning policies for exploration. The closest ones are Gal & Ghahramani (2016); Henderson et al. (2017a), which use dropout in value network to capture agents' uncertainty about the environment. According to Osband et al. (2016), Gal & Ghahramani (2016) even fails in environment requiring temporally consistent exploration.

There are also some attempts from the Evolutionary Strategies and Genetic Algorithms literature (Salimans et al., 2017; Such et al., 2017; Gangwani & Peng, 2017) to continuous control tasks. Though they model the problem much differently from ours, the relation could be an interesting topic for future research.

## 6 CONCLUSION

Building a hierarchy of stochasticity for reinforcement learning policy is the first step towards more structured exploration. We presented a method, NADPEx, that models stochasticity at large time scale with a distribution of plausible subnetworks from the same complete network to achieve on-policy temporally consistent exploration. These subnetworks are sampled through dropout at the beginning of episodes, used to explore the environment with diverse and consistent behavioral patterns and updated through simultaneous gradient back-propagation. A learning objective is provided such that this distribution is also updated in an *end-to-end* manner to adapt to the action-space policy. Thanks to the fact that this dropout transformation is differentiable, KL regularizers on policy space can help to further stabilize it. Our experiments exhibit that NADPEx successfully solves continuous control tasks, even with strong sparsity in rewards.

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

## A    KL DIVERGENCE FIRST ORDER APPROXIMATION

We derive the first order derivatitve of KL divergence from a refernce policy $\overline{\pi}$ to a parametrized policy $\pi_{\boldsymbol{\theta}}$ to :

$$
\begin{aligned}
\nabla_{\boldsymbol{\theta}} D_{KL}(\pi_{\boldsymbol{\theta}}(\cdot|s)||\overline{\pi}(\cdot|s)) &= \nabla_{\boldsymbol{\theta}} \int \pi_{\boldsymbol{\theta}}(a|s)(\log \pi_{\boldsymbol{\theta}}(a|s) - \log \overline{\pi}(a|s))da \\
&= \int \nabla_{\boldsymbol{\theta}}(\pi_{\boldsymbol{\theta}}(a|s)(\log \pi_{\boldsymbol{\theta}}(a|s) - \log \overline{\pi}(a|s)))da \\
&= \int \nabla_{\boldsymbol{\theta}}\pi_{\boldsymbol{\theta}}(a|s)(\log \pi_{\boldsymbol{\theta}}(a|s) - \log \overline{\pi}(a|s) + 1)da \\
&= \int \pi_{\boldsymbol{\theta}}(a|s)\nabla_{\boldsymbol{\theta}} \log \pi_{\boldsymbol{\theta}}(a|s)(\log \pi_{\boldsymbol{\theta}}(a|s) - \log \overline{\pi}(a|s) + 1)da \\
&= \mathbb{E}_{a\sim\pi_{\boldsymbol{\theta}}}[\nabla_{\boldsymbol{\theta}} \log \pi_{\boldsymbol{\theta}}(a|s)(\log \pi_{\boldsymbol{\theta}}(a|s) - \log \overline{\pi}(a|s))]
\end{aligned}
\tag{13}
$$

Note that line 3 derives line 4 with likelihood ratio trick, line 4 derives line 5 as $\int \pi_{\boldsymbol{\theta}}(a|s)\nabla_{\boldsymbol{\theta}} \log \pi_{\boldsymbol{\theta}}(a|s)da = 0$

## B    GRADIENTS OF NADPEX

Here we provide a full derivation of NADPEx's gradients. Specifically, gradients for two types of dropout are discussed.

As shown in (7), a NADPEx policy $\pi_{\boldsymbol{\theta},\boldsymbol{z}}$ is a joint distribution, which could be factorized with a *dropout distribution* $q_{\phi}(\boldsymbol{z})$ and a conditional distribution, *i.e.* the *dropout policy*, $\pi_{\boldsymbol{\theta}|\boldsymbol{z}}$, $\boldsymbol{z} \sim q_{\phi}(\boldsymbol{z})$ is the dropout random variable vector.

In reinforcement learning with NADPEx, we use gradient based optimization to maximize the objective (8).

$$
\nabla\eta(\pi_{\boldsymbol{\theta},\phi}) = \nabla\mathbb{E}_{\mathbf{z}\sim q(\mathbf{z})}[\mathbb{E}_{\tau|\boldsymbol{z}}[\sum_{t=0}^{T} \gamma^t r(s_t, a_t)]].
\tag{14}
$$

### B.1    DISCRETE DROPOUT

Normally the *dropout distribution* $q_{\phi}(\boldsymbol{z})$ is a discrete distribution, for example, $\boldsymbol{z} \sim Bernoulli(\boldsymbol{\phi})$, ones can use likelihood ratio trick to calculate (14):

$$
\begin{aligned}
\nabla_{\boldsymbol{\theta},\phi}\eta(\pi_{\boldsymbol{\theta},\phi}) &= \nabla_{\boldsymbol{\theta},\phi}\mathbb{E}_{\mathbf{z}\sim q_{\phi}}[\mathbb{E}_{\tau|\boldsymbol{z}}[\sum_{t=0}^{T} \gamma^t r(s_t, a_t)]] \\
&= \nabla_{\boldsymbol{\theta},\phi} \int q_{\phi}(\boldsymbol{z}) \int p_{\tau|\boldsymbol{z},\boldsymbol{\theta}} R_{\tau} d\tau d\boldsymbol{z} \\
&= \int q_{\phi}(\boldsymbol{z})\nabla_{\boldsymbol{\theta}} \int p_{\tau|\boldsymbol{z},\boldsymbol{\theta}} R_{\tau} d\tau d\boldsymbol{z} + \nabla_{\phi} \int q_{\phi}(\boldsymbol{z}) \int p_{\tau|\boldsymbol{z},\boldsymbol{\theta}} R_{\tau} d\tau d\boldsymbol{z} \\
&= \int q_{\phi}(\boldsymbol{z})\nabla_{\boldsymbol{\theta}} \int p_{\tau|\boldsymbol{z},\boldsymbol{\theta}} R_{\tau} d\tau d\boldsymbol{z} + \nabla_{\phi} \int q_{\phi}(\boldsymbol{z})\mathbb{E}_{\tau|\boldsymbol{z}}[R(s_{0|\boldsymbol{z}}, a_{0|\boldsymbol{z}})]d\boldsymbol{z} \\
&= \mathbb{E}_{\mathbf{z}\sim q_{\phi}}[\nabla_{\boldsymbol{\theta}}\mathcal{L}(\boldsymbol{\theta},\boldsymbol{z}) + \nabla_{\phi} \log q_{\phi}(\boldsymbol{z})A(s_{0|\boldsymbol{z}}, a_{0|\boldsymbol{z}})] \\
&\approx \frac{1}{N}\sum_{i=1}^{N}(\nabla_{\boldsymbol{\theta}}\mathcal{L}(\boldsymbol{\theta},\boldsymbol{z}^i) + \nabla_{\phi} \log q_{\phi}(\boldsymbol{z}^i)A(s_0^i, a_0^i)),
\end{aligned}
\tag{15}
$$

where $\mathcal{L}(\cdot)$ is the surrogate loss, varying from reinforcement learning algorihtms, $A(s_0^i, a_0^i)$ is the GAE (Schulman et al., 2015c) for the $i$th trajectory from the beginning *i.e.* $(s_0^i, a_0^i)$, such that the gradient has low variance.

### B.2 Gaussian multiplicative dropout

In Srivastava et al. (2014), *Gaussian multiplicative dropout* is proposed, where $\mathbf{z} \sim \mathcal{N}(\mathbf{I}, \boldsymbol{\sigma}^2)$ is the multiplicative dropout random variable vector, with the same expectation of *discrete dropout* $P(z = 1) = \frac{1}{1+\phi}$. Reparametirzed with a dummy random variable $\boldsymbol{\epsilon} \sim \mathcal{N}(\mathbf{0}, \mathbf{I})$ (Kingma & Welling, 2013; Rezende et al., 2014), we have $\boldsymbol{z} = \mathbf{I} + \boldsymbol{\phi} \odot \boldsymbol{\epsilon}$, where $\boldsymbol{\phi}$ is used to denote $\boldsymbol{\sigma}$ for consistency of notation, $\odot$ is an element-wise multiplication, such that (14) could be calculated as:

$$
\begin{aligned}
\nabla_{\boldsymbol{\theta},\boldsymbol{\phi}}\eta(\pi_{\boldsymbol{\theta},\boldsymbol{\phi}}) &= \nabla_{\boldsymbol{\theta},\boldsymbol{\phi}}\mathbb{E}_{\mathbf{z}\sim\mathcal{N}(\boldsymbol{I},\boldsymbol{\phi}^2)}[\mathbb{E}_{\tau|\mathbf{z}}[\sum_{t=0}^{T}\gamma^t r(s_t, a_t)]] \\
&= \nabla_{\boldsymbol{\theta},\boldsymbol{\phi}}\mathbb{E}_{\mathbf{z}\sim\mathcal{N}(\boldsymbol{I},\boldsymbol{\phi}^2)}[\eta(\pi_{\boldsymbol{\theta}|\mathbf{z}})] \\
&= \nabla_{\boldsymbol{\theta},\boldsymbol{\phi}}\mathbb{E}_{\boldsymbol{\epsilon}\sim\mathcal{N}(\boldsymbol{0},\boldsymbol{I})}[\eta(\pi_{\boldsymbol{\theta}|\mathbf{I}+\boldsymbol{\phi}\odot\boldsymbol{\epsilon}})] \\
&= \mathbb{E}_{\boldsymbol{\epsilon}\sim\mathcal{N}(\boldsymbol{0},\boldsymbol{I})}[\nabla_{\boldsymbol{\theta},\boldsymbol{\phi}}\eta(\pi_{\boldsymbol{\theta}|\mathbf{I}+\boldsymbol{\phi}\odot\boldsymbol{\epsilon}})] \\
&= \mathbb{E}_{\boldsymbol{\epsilon}\sim\mathcal{N}(\boldsymbol{0},\boldsymbol{I})}[\nabla_{\boldsymbol{\theta},\boldsymbol{\phi}}\mathcal{L}(\boldsymbol{\theta}, \mathbf{I} + \boldsymbol{\phi} \odot \boldsymbol{\epsilon})] \\
&\approx \frac{1}{N}\sum_{i=1}^{N}\nabla_{\boldsymbol{\theta},\boldsymbol{\phi}}\mathcal{L}(\boldsymbol{\theta}, \mathbf{I} + \boldsymbol{\phi} \odot \boldsymbol{\epsilon}^i)
\end{aligned}
\tag{16}
$$

## C  NADPEx KL divergence first order approximation

In NADPEx, the *dropout distribution* is designed to be fully factorizable, thus the gradient of KL divergence in the NADPEx policy space is obtainable:

$$
\begin{aligned}
&\nabla_{\boldsymbol{\theta},\boldsymbol{\phi}}D_{KL}(\pi_{\boldsymbol{\theta},\boldsymbol{\phi}}(\cdot|\boldsymbol{s})||\pi_{\boldsymbol{\theta}^{old},\boldsymbol{\phi}^{old}}(\cdot|\boldsymbol{s})) \\
=&\nabla_{\boldsymbol{\theta},\boldsymbol{\phi}}\int\int\pi_{\boldsymbol{\theta},\boldsymbol{\phi}}(\boldsymbol{a},\boldsymbol{z}|\boldsymbol{s})\log\frac{\pi_{\boldsymbol{\theta},\boldsymbol{\phi}}(\boldsymbol{a},\boldsymbol{z}|\boldsymbol{s})}{\pi_{\boldsymbol{\theta}^{old},\boldsymbol{\phi}^{old}}(\boldsymbol{a},\boldsymbol{z}|\boldsymbol{s})}d\boldsymbol{a}d\boldsymbol{z} \\
=&\nabla_{\boldsymbol{\theta},\boldsymbol{\phi}}\int\int q_{\boldsymbol{\phi}}(\boldsymbol{z})\pi_{\boldsymbol{\theta}|\boldsymbol{z}}(\boldsymbol{a}|\boldsymbol{s})\log\frac{q_{\boldsymbol{\phi}}(\boldsymbol{z})\pi_{\boldsymbol{\theta}|\boldsymbol{z}}(\boldsymbol{a}|\boldsymbol{s})}{q_{\boldsymbol{\phi}^{old}}(\boldsymbol{z})\pi_{\boldsymbol{\theta}^{old}|\boldsymbol{z}}(\boldsymbol{a}|\boldsymbol{s})}d\boldsymbol{a}d\boldsymbol{z} \\
=&\int q_{\boldsymbol{\phi}}(\boldsymbol{z})\nabla_{\boldsymbol{\theta}}\int\pi_{\boldsymbol{\theta}|\boldsymbol{z}}(\boldsymbol{a}|\boldsymbol{s})\log\frac{q_{\boldsymbol{\phi}}(\boldsymbol{z})\pi_{\boldsymbol{\theta}|\boldsymbol{z}}(\boldsymbol{a}|\boldsymbol{s})}{q_{\boldsymbol{\phi}^{old}}(\boldsymbol{z})\pi_{\boldsymbol{\theta}^{old}|\boldsymbol{z}}(\boldsymbol{a}|\boldsymbol{s})}d\boldsymbol{a}d\boldsymbol{z} \\
&+\nabla_{\boldsymbol{\phi}}\int q_{\boldsymbol{\phi}}(\boldsymbol{z})\int\pi_{\boldsymbol{\theta}|\boldsymbol{z}}(\boldsymbol{a}|\boldsymbol{s})\log\frac{q_{\boldsymbol{\phi}}(\boldsymbol{z})\pi_{\boldsymbol{\theta}|\boldsymbol{z}}(\boldsymbol{a}|\boldsymbol{s})}{q_{\boldsymbol{\phi}^{old}}(\boldsymbol{z})\pi_{\boldsymbol{\theta}^{old}|\boldsymbol{z}}(\boldsymbol{a}|\boldsymbol{s})}d\boldsymbol{a}d\boldsymbol{z}.
\end{aligned}
\tag{17}
$$

And thus the same first-order approximation could be done just as in Appendix A. However, note that likelihood ratio trick is used during the derivation, which should be applied to the random variable vectors $\boldsymbol{a}$ and $\boldsymbol{z}$ here.

### C.1  Reducing variance in Monte Carlo estimate with MuProp

As normally $\boldsymbol{z}$ could have a much higher dimension than $\boldsymbol{a}$, making this likelihood ratio trick suffer from *curse of dimensionality*, we inspect into the second term with MuProp (Gu et al., 2016) for a low-variance estimate. Let

$$
f(\boldsymbol{z}) = \int\pi_{\boldsymbol{\theta}|\boldsymbol{z}}(\boldsymbol{a}|\boldsymbol{s})\log\frac{q_{\boldsymbol{\phi}}(\boldsymbol{z})\pi_{\boldsymbol{\theta}|\boldsymbol{z}}(\boldsymbol{a}|\boldsymbol{s})}{q_{\boldsymbol{\phi}^{old}}(\boldsymbol{z})\pi_{\boldsymbol{\theta}^{old}|\boldsymbol{z}}(\boldsymbol{a}|\boldsymbol{s})}d\boldsymbol{a},
\tag{18}
$$

then we have

$$
\begin{aligned}
g_{\boldsymbol{\phi}} &= \nabla_{\boldsymbol{\phi}}\int q_{\boldsymbol{\phi}}(\boldsymbol{z})f(\boldsymbol{z})d\boldsymbol{z} = \int q_{\boldsymbol{\phi}}\nabla_{\boldsymbol{\phi}}\log q_{\boldsymbol{\phi}}(\boldsymbol{z})f(\boldsymbol{z})d\boldsymbol{z} \\
&= \mathbb{E}_{\boldsymbol{z}}[\nabla_{\boldsymbol{\phi}}\log q_{\boldsymbol{\phi}}(\boldsymbol{z})f(\boldsymbol{z})] \\
\hat{g}_{\boldsymbol{\phi}} &= \nabla_{\boldsymbol{\phi}}\log q_{\boldsymbol{\phi}}(\boldsymbol{z})f(\boldsymbol{z}) \quad where \quad z \sim q_{\boldsymbol{\phi}}.
\end{aligned}
\tag{19}
$$

The idea of MuProp is to use a control variate that corresponds to the first-order Taylor expansion of $f$ around some fixed value $\bar{z}$, *i.e.* $h(z) = f(\bar{z}) + f'(\bar{z})(z - \bar{z})$, such that

$$
\begin{aligned}
g_{\boldsymbol{\phi}}^M &= \nabla_{\boldsymbol{\phi}} \mathbb{E}_{\boldsymbol{z}}[(f(\boldsymbol{z}) - h(\boldsymbol{z})) + h(\boldsymbol{z})] \\
&= \nabla_{\boldsymbol{\phi}} \mathbb{E}_{\boldsymbol{z}}[f(\boldsymbol{z}) - f(\bar{\boldsymbol{z}}) - f'(\bar{\boldsymbol{z}})(\boldsymbol{z} - \bar{\boldsymbol{z}})] + (f(\bar{\boldsymbol{z}}) + f'(\bar{\boldsymbol{z}})(\boldsymbol{z} - \bar{\boldsymbol{z}}))] \\
&= \mathbb{E}_{\boldsymbol{z}}[\nabla_{\boldsymbol{\phi}} \log q_{\boldsymbol{\phi}}(\boldsymbol{z})[f(\boldsymbol{z}) - h(\boldsymbol{z})]] + f'(\bar{\boldsymbol{z}}) \nabla_{\boldsymbol{\phi}} \mathbb{E}_{\boldsymbol{z}}[\boldsymbol{z}] \\
\hat{g}_{\boldsymbol{\phi}}^M &= \nabla_{\boldsymbol{\phi}} \log q_{\boldsymbol{\phi}}(\boldsymbol{z})[f(\boldsymbol{z}) - h(\boldsymbol{z})] + f'(\bar{\boldsymbol{z}}) \nabla_{\boldsymbol{\phi}} \mathbb{E}_{\boldsymbol{z}}[\boldsymbol{z}] \quad where \quad z \sim q_{\boldsymbol{\phi}}.
\end{aligned}
\tag{20}
$$

To further reduce the variance, the first term is sometimes omitted with acceptable biased introduced (Raiko et al., 2015). And empirically, we find it could be almost negligible comparing with the gradient from the reinforcement learning objective. For *Gaussian dropout*, $\nabla_{\boldsymbol{\phi}} \mathbb{E}_{\boldsymbol{z}}[\boldsymbol{z}] = \nabla_{\boldsymbol{\phi}} \mathbf{1} = \mathbf{0}$, $g_{\boldsymbol{\phi}}$ is thus eliminated. With *binary dropout*, $\nabla_{\boldsymbol{\phi}} \mathbb{E}_{\boldsymbol{z}}[\boldsymbol{z}] = \nabla_{\boldsymbol{\phi}} \boldsymbol{\phi} = \mathbf{1}$, we have

$$
\begin{aligned}
\hat{g}_{\boldsymbol{\phi}}^M &= f'(\boldsymbol{z})|_{\boldsymbol{z}=\bar{\boldsymbol{z}}} \\
&= \frac{\partial}{\partial \boldsymbol{z}} \log \frac{q_{\boldsymbol{\phi}}(\boldsymbol{z})}{q_{\boldsymbol{\phi}^{old}}(\boldsymbol{z})} + \frac{\partial}{\partial \boldsymbol{z}} \int \pi_{\boldsymbol{\theta}|\boldsymbol{z}}(\boldsymbol{a}|\boldsymbol{s}) \log \frac{\pi_{\boldsymbol{\theta}|\boldsymbol{z}}(\boldsymbol{a}|\boldsymbol{s})}{\pi_{\boldsymbol{\theta}^{old}|\boldsymbol{z}}(\boldsymbol{a}|\boldsymbol{s})} d\boldsymbol{a} \\
&= \frac{\partial}{\partial \boldsymbol{z}} \log \frac{\prod_{i=0}^{m-1} \boldsymbol{\phi}^{\boldsymbol{z}_i} (1 - \boldsymbol{\phi}_i)^{1-\boldsymbol{z}_i}}{\prod_{i=0}^{m-1} \boldsymbol{\phi}_i^{old\,\boldsymbol{z}_i} (1 - \boldsymbol{\phi}_i^{old})^{1-\boldsymbol{z}_i}} + \frac{\partial}{\partial \boldsymbol{z}} \int \pi_{\boldsymbol{\theta}|\boldsymbol{z}}(\boldsymbol{a}|\boldsymbol{s}) \log \frac{\pi_{\boldsymbol{\theta}|\boldsymbol{z}}(\boldsymbol{a}|\boldsymbol{s})}{\pi_{\boldsymbol{\theta}^{old}|\boldsymbol{z}}(\boldsymbol{a}|\boldsymbol{s})} d\boldsymbol{a} \\
&= \sum_{i=0}^{m-1} \log \frac{\boldsymbol{\phi}_i (1 - \boldsymbol{\phi}_i^{old})}{(1 - \boldsymbol{\phi}_i) \boldsymbol{\phi}_i^{old}} + \frac{\partial}{\partial \boldsymbol{z}} \int \pi_{\boldsymbol{\theta}|\boldsymbol{z}}(\boldsymbol{a}|\boldsymbol{s}) \log \frac{\pi_{\boldsymbol{\theta}|\boldsymbol{z}}(\boldsymbol{a}|\boldsymbol{s})}{\pi_{\boldsymbol{\theta}^{old}|\boldsymbol{z}}(\boldsymbol{a}|\boldsymbol{s})} d\boldsymbol{a}.
\end{aligned}
\tag{21}
$$

Note that in line 3 we assume the probabilistic distribution function to be continuous in an infinitesimal regions around $\{\boldsymbol{z}|\boldsymbol{z}_i \in \{0, 1\} \quad for \quad \forall \boldsymbol{z}_i\}$ to allow the existence of the first order derivative for the first term. The first term is then close to zero as $\boldsymbol{\phi}$ is close to $\boldsymbol{\phi}^{old}$. Another possible relaxation method is to use *Concrete distribution* (Maddison et al., 2016):

$$
q_{\boldsymbol{\phi},\lambda}(\boldsymbol{z}) = \prod_{i=0}^{m-1} \frac{\lambda \boldsymbol{\phi}_i \boldsymbol{z}_i^{-\lambda_i - 1} (1 - \boldsymbol{z}_i)^{-\lambda - 1}}{(\boldsymbol{\phi}_i \boldsymbol{z}_i^{-\lambda} + (1 - \boldsymbol{z}_i)^{-\lambda})^2},
\tag{22}
$$

where $\lambda$ is the temperature of this relaxation. Substituting it back to (21), we have

$$
\begin{aligned}
\hat{g}_{\boldsymbol{\phi}}^M &= f'(\boldsymbol{z})|_{\boldsymbol{z}=\bar{\boldsymbol{z}}} \\
&= \frac{\partial}{\partial \boldsymbol{z}} \log \frac{q_{\boldsymbol{\phi},\lambda}(\boldsymbol{z})}{q_{\boldsymbol{\phi}^{old},\lambda}(\boldsymbol{z})} + \frac{\partial}{\partial \boldsymbol{z}} \int \pi_{\boldsymbol{\theta}|\boldsymbol{z}}(\boldsymbol{a}|\boldsymbol{s}) \log \frac{\pi_{\boldsymbol{\theta}|\boldsymbol{z}}(\boldsymbol{a}|\boldsymbol{s})}{\pi_{\boldsymbol{\theta}^{old}|\boldsymbol{z}}(\boldsymbol{a}|\boldsymbol{s})} d\boldsymbol{a} \\
&= \frac{\partial}{\partial \boldsymbol{z}} \sum_{i=0}^{m-1} (\log \frac{\lambda \boldsymbol{\phi}_i}{\lambda \boldsymbol{\phi}_i^{old}} - 2 \log(\frac{\boldsymbol{\phi}_i \boldsymbol{z}_i^{-\lambda} + (1 - \boldsymbol{z}_i)^{-\lambda}}{\boldsymbol{\phi}_i^{old} \boldsymbol{z}_i^{-\lambda} + (1 - \boldsymbol{z}_i)^{-\lambda}})) + \frac{\partial}{\partial \boldsymbol{z}} \int \pi_{\boldsymbol{\theta}|\boldsymbol{z}}(\boldsymbol{a}|\boldsymbol{s}) \log \frac{\pi_{\boldsymbol{\theta}|\boldsymbol{z}}(\boldsymbol{a}|\boldsymbol{s})}{\pi_{\boldsymbol{\theta}^{old}|\boldsymbol{z}}(\boldsymbol{a}|\boldsymbol{s})} d\boldsymbol{a} \\
&= -2 \sum_{i=0}^{m-1} \frac{\partial}{\partial \boldsymbol{z}} (\log(\frac{\boldsymbol{\phi}_i \boldsymbol{z}_i^{-\lambda} + (1 - \boldsymbol{z}_i)^{-\lambda}}{\boldsymbol{\phi}_i^{old} \boldsymbol{z}_i^{-\lambda} + (1 - \boldsymbol{z}_i)^{-\lambda}})) + \frac{\partial}{\partial \boldsymbol{z}} \int \pi_{\boldsymbol{\theta}|\boldsymbol{z}}(\boldsymbol{a}|\boldsymbol{s}) \log \frac{\pi_{\boldsymbol{\theta}|\boldsymbol{z}}(\boldsymbol{a}|\boldsymbol{s})}{\pi_{\boldsymbol{\theta}^{old}|\boldsymbol{z}}(\boldsymbol{a}|\boldsymbol{s})} d\boldsymbol{a} \\
&= 2\lambda \sum_{i=0}^{m-1} (\frac{\boldsymbol{\phi}_i \boldsymbol{z}_i^{-\lambda-1} + (1 - \boldsymbol{z}_i)^{-\lambda-1}}{\boldsymbol{\phi}_i \boldsymbol{z}_i^{-\lambda} + (1 - \boldsymbol{z}_i)^{-\lambda}} - \frac{\boldsymbol{\phi}_i^{old} \boldsymbol{z}_i^{-\lambda-1} + (1 - \boldsymbol{z}_i)^{-\lambda-1}}{\boldsymbol{\phi}_i^{old} \boldsymbol{z}_i^{-\lambda} + (1 - \boldsymbol{z}_i)^{-\lambda}}) \\
&+ \frac{\partial}{\partial \boldsymbol{z}} \int \pi_{\boldsymbol{\theta}|\boldsymbol{z}}(\boldsymbol{a}|\boldsymbol{s}) \log \frac{\pi_{\boldsymbol{\theta}|\boldsymbol{z}}(\boldsymbol{a}|\boldsymbol{s})}{\pi_{\boldsymbol{\theta}^{old}|\boldsymbol{z}}(\boldsymbol{a}|\boldsymbol{s})} d\boldsymbol{a}
\end{aligned}
\tag{23}
$$

We can reach the same claim as above that the first term could be removed if we set $\boldsymbol{z} = \bar{\boldsymbol{z}} \to 0$. And the second term could also be eliminated as first order derivative of KL divergence between two different distributions parametrized with almost but not entirely null networks if $\bar{\boldsymbol{z}} \to 0$.

## C.2 RELAXING ANALYTICAL KL DIVERGENCE WITH TRUST REGION

Ones may argue that the second term in (17) is decomposable to

$$
\nabla_{\boldsymbol{\phi}} \int q_{\boldsymbol{\phi}}(\boldsymbol{z})(\log q_{\boldsymbol{\phi}}(\boldsymbol{z}) - \log q_{\boldsymbol{\phi}^{old}}(\boldsymbol{z})) d\boldsymbol{z} = \nabla_{\boldsymbol{\phi}} D_{KL}(q_{\boldsymbol{\phi}_i}(\boldsymbol{z}_i) || q_{\boldsymbol{\phi}_i^{old}}(\boldsymbol{z}_i)),
\tag{24}
$$

$$\nabla_{\boldsymbol{\phi}} \int q_{\boldsymbol{\phi}}(\boldsymbol{z}) \int \pi_{\boldsymbol{\theta}|\boldsymbol{z}}(\boldsymbol{a}|\boldsymbol{s})(\log \pi_{\boldsymbol{\theta}|\boldsymbol{z}}(\boldsymbol{a}|\boldsymbol{s}) - \log \pi_{\boldsymbol{\theta}^{old}|\boldsymbol{z}}(\boldsymbol{a}|\boldsymbol{s}))d\boldsymbol{a}d\boldsymbol{z} \tag{25}$$

such that (24) has analytical form as $q$ is either diagonal Gaussian or Bernoulli and thus the Monte Carlo gradient estimate and the MuProp approximation above need only to be exerted on (25). However, it can be proved that $\nabla_{\boldsymbol{\phi}} D_{KL}(q_{\boldsymbol{\phi}_i}(\boldsymbol{z}_i)||q_{\boldsymbol{\phi}_i^{old}}(\boldsymbol{z}_i))$ is still omittable with *trust region* (Schulman et al., 2015b). When first proposed in Schulman et al. (2015b), *trust region* is a relaxation on $D_{KL}$ to encourage efficient learning, with the idea that step size will only be adapted if the *trust region* constraint is violated, *i.e.* $D_{KL} > \delta$. The adaptation of step size is replaced in Wang et al. (2016) with a mechanism to clip gradients from surrogate loss adaptively only if $D_{KL} > \delta$ to further boost the efficiency.

We prove below that $D_{kl}$ will almost never violate the constraint $\delta = 1$ proposed in Wang et al. (2016). For *Gaussian dropout* we have:

$$
\begin{aligned}
D_{KL}(q_{\boldsymbol{\phi}_i}(\boldsymbol{z}_i)||q_{\boldsymbol{\phi}_i^{old}}(\boldsymbol{z}_i)) &= \log \frac{\phi_i^{old}}{\phi_i} + \frac{\phi_i^2}{2\phi_i^{old2}} - \frac{1}{2} = \log \frac{\phi_i^{old}}{\phi_i^{old} + \Delta\phi_i} + \frac{(\phi_i^{old} + \Delta\phi_i)^2}{2\phi_i^{old2}} - \frac{1}{2} \\
&= \log \frac{1}{1 + \frac{\Delta\phi_i}{\phi_i^{old}}} + \frac{\Delta\phi_i}{\phi_i^{old}} + \frac{\Delta\phi_i^2}{\phi_i^{old2}} = -\log(1 + \frac{\Delta\phi_i}{\phi_i^{old}}) + \frac{\Delta\phi_i}{\phi_i^{old}} + \frac{\Delta\phi_i^2}{\phi_i^{old2}} \\
&= -\log(1 + x) + x + x^2,
\end{aligned}
\tag{26}
$$

where we let $x = \frac{\Delta\phi_i}{\phi_i^{old}}$. According to our experiment, as well as some conclusions in the literature for dropout (Kingma et al., 2015), $\phi_i \in (0.005, 0.5)$. Obviously $x \in (0, 1)$, $D_{KL}(q_{\boldsymbol{\phi}_i}(\boldsymbol{z}_i)||q_{\boldsymbol{\phi}_i^{old}}(\boldsymbol{z}_i)) \in (0, 1)$.

For *binary dropout* we have:

$$
\begin{aligned}
&D_{KL}(q_{\boldsymbol{\phi}_i}(\boldsymbol{z}_i)||q_{\boldsymbol{\phi}_i^{old}}(\boldsymbol{z}_i)) \\
=&\phi_i(\log \phi_i - \log \phi_i^{old}) + (1 - \phi_i)(\log(1 - \phi_i) - \log(1 - \phi_i^{old})) \\
=&[\log(1 - \phi_i) - \log(1 - \phi_i^{old})] + \phi_i[\log \frac{\phi_i}{1 - \phi_i} - \log \frac{\phi_i^{old}}{1 - \phi_i^{old}}] \\
=&[\log(1 - \phi_i^{old} - \Delta\phi_i) - \log(1 - \phi_i^{old})] + \phi_i[\log \frac{\phi_i^{old} + \Delta\phi_i}{1 - \phi_i^{old} - \Delta\phi_i} - \log \frac{\phi_i^{old}}{1 - \phi_i^{old}}].
\end{aligned}
\tag{27}
$$

We can thus discuss how $D_{KL}$ changes with $\Delta\phi_i$ and $\phi_i^{old}$:

$$
\begin{aligned}
&\frac{\partial}{\partial \Delta\phi_i} D_{KL}(q_{\boldsymbol{\phi}_i}(\boldsymbol{z}_i)||q_{\boldsymbol{\phi}_i^{old}}(\boldsymbol{z}_i)) \\
=& -\frac{1}{1 - \phi_i^{old} - \Delta\phi_i} + (\phi_i^{old} + \Delta\phi_i)(\frac{1}{\phi_i^{old} + \Delta\phi_i} + \frac{1}{1 - \phi_i^{old} - \Delta\phi_i}) \\
&+ (\log \frac{\phi_i^{old} + \Delta\phi_i}{1 - \phi_i^{old} - \Delta\phi_i} - \log \frac{\phi_i^{old}}{1 - \phi_i^{old}}) \\
=& \log \frac{\phi_i^{old} + \Delta\phi_i}{1 - \phi_i^{old} - \Delta\phi_i} - \log \frac{\phi_i^{old}}{1 - \phi_i^{old}} = h(\phi_i^{old} + \Delta\phi_i) - h(\phi_i^{old}).
\end{aligned}
\tag{28}
$$

$$
\begin{aligned}
&\frac{\partial}{\partial \phi_i^{old}} D_{KL}(q_{\boldsymbol{\phi}_i}(\boldsymbol{z}_i)||q_{\boldsymbol{\phi}_i^{old}}(\boldsymbol{z}_i)) \\
=& -\frac{1}{1 - \phi_i^{old} - \Delta\phi_i} + \frac{1}{1 - \phi_i^{old}} + (\phi_i^{old} + \Delta\phi_i)(\frac{1}{\phi_i^{old} + \Delta\phi_i} + \frac{1}{1 - \phi_i^{old} - \Delta\phi_i}) \\
&+ (\phi_i^{old} + \Delta\phi_i)(\frac{1}{\phi_i^{old}} + \frac{1}{1 - \phi_i^{old}}) + (\log \frac{\phi_i^{old} + \Delta\phi_i}{1 - \phi_i^{old} - \Delta\phi_i} - \log \frac{\phi_i^{old}}{1 - \phi_i^{old}}) \\
=& -\Delta\phi_i(\frac{1}{1 - \phi_i^{old}} + \frac{1}{\phi_i^{old}}) + \log \frac{\phi_i^{old} + \Delta\phi_i}{1 - \phi_i^{old} - \Delta\phi_i} - \log \frac{\phi_i^{old}}{1 - \phi_i^{old}} \\
=& \frac{1}{2!} h''(\phi_i^{old})\Delta\phi_i + \frac{1}{3!} h'''(\phi_i^{old})\Delta\phi_i^2 + ...
\end{aligned}
\tag{29}
$$

With $h(x) = \log \frac{x}{1-x}, x \in (0.005, 0.5)$, it is easy to see that $\frac{\partial D_{KL}}{\partial \Delta \phi_i} = 0$ when $\Delta \phi_i = 0$, $\frac{\partial D_{KL}}{\partial \Delta \phi_i} > 0$ when $\Delta \phi_i > 0$ and $\frac{\partial D_{KL}}{\partial \Delta \phi_i} < 0$ when $\Delta \phi_i < 0$. Similarly, when $\phi^{old} = 0.5$, $\frac{\partial D_{KL}}{\partial \phi^{old}} = 0$; when $\phi^{old} > 0.5$, $\frac{\partial D_{KL}}{\partial \phi^{old}} > 0$; when $\phi^{old} < 0.5$, $\frac{\partial D_{KL}}{\partial \phi^{old}} < 0$. Hence $D_{KL}(q_{\phi_i}(z_i)||q_{\phi_i^{old}}(z_i))$ reaches its maximum at $|\Delta \phi_i|_{max}$ and $|\phi_i^{old} - 0.5|_{max}$. It is reasonable to set $|\Delta \phi_i|_{max} = 0.1$, leading us to $D_{KL}(q_{\phi_i}(z_i)||q_{\phi_i^{old}}(z_i))_{max} \approx 0.225 < \delta = 1$. We reach the same claim that $\nabla_\phi D_{KL}(\pi_{\theta,\phi}(\cdot|s)||\pi_{\theta^{old},\phi^{old}}(\cdot|s))$ could be stopped for the sake of learning efficiency at a cost of acceptable bias.

## C.3 REMEDY BIAS WITH MEAN POLICY

As derived above, the regularization term only back-propagates gradients to $\theta$:

$$
\begin{aligned}
&\nabla_\theta D_{KL}(\pi_{\theta,\phi}(\cdot|s)||\pi_{\theta^{old},\phi^{old}}(\cdot|s)) \\
&= \int q_\phi(z) \nabla_\theta \int \pi_{\theta|z}(a|s) \log \frac{q_\phi(z)\pi_{\theta|z}(a|s)}{q_{\phi^{old}}(z)\pi_{\theta^{old}|z}(a|s)} da dz \\
&= \int q_\phi(z) \nabla_\theta \int \pi_{\theta|z}(a|s)(\log \frac{\pi_{\theta|z}(a|s)}{\pi_{\theta^{old}|z}(a|s)} + \log \frac{q_\phi(z)}{q_{\phi^{old}}(z)}) da dz \\
&= \int q_\phi(z) \nabla_\theta \int \pi_{\theta|z}(a|s)(\log \pi_{\theta|z}(a|s) - \log \pi_{\theta^{old}|z}(a|s)) da dz \\
&= \mathbb{E}_{z \sim q_\phi}[\nabla_\theta D_{KL}(\pi_{\theta|z}(\cdot|s)||\pi_{\theta^{old}|z}(\cdot|s))] \\
&\approx \frac{1}{N} \sum_{i=0}^{N-1} \nabla_\theta D_{KL}(\pi_{\theta|z_i}(\cdot|s)||\pi_{\theta^{old}|z_i}(\cdot|s)),
\end{aligned}
\tag{30}
$$

where $N$ is the number of *dropout policies* in this batch. Note that from line 3 to line 4 we simply remove $\log \frac{q_\phi(z)}{q_{\phi^{old}}(z)}$, which can be regarded as substracting it as a baseline $b(z)$ to reduce variance. Obviously, (30) has similar level of variance as (9), closing our discussion on variance reduction even though some other techniques could possibly go further.

Seeing the lack of regularization on $\phi$ due to the gradient omission, we further enforce the idea that *dropout policy* had better to be close to each other, which is the supposed role of regularizer on $\phi$. This could be intuitively understood as a remedy from $\theta$ to $\phi$:

$$
\frac{1}{N} \sum_{i=0}^{N-1} \sum_{j=0}^{N-1} \nabla_\theta D_{KL}(\pi_{\theta|z_i}(\cdot|s)||\pi_{\theta^{old}|z_j}(\cdot|s)).
\tag{31}
$$

Noticing that this term has a complexity of $O(N^2)$, we replace $\pi_{\theta,\phi}$ with a *mean policy* $\pi_\theta = \pi_{\theta|\bar{z}}$. In the deep learning literature, it is fairly prevalent to use *mean network* for dropout training with some adjustment in gradient back-propagation, due to stability and effiency concern. For instance, it is proved by Wang & Manning (2013) that the sampling process for *dropout network* with *sigmoid* or *ReLU* activators could be avoided by integrating a Gaussian approximation at each layer of a neural network. Therefore, we have

$$
\begin{aligned}
&\frac{1}{N} \sum_{i=0}^{N-1} \nabla_\theta D_{KL}(\pi_\theta(\cdot|s)||\pi_{\theta^{old}|z_i}(\cdot|s)) \\
&= \frac{1}{N} \sum_{i=0}^{N-1} \nabla_\theta \int \pi_\theta(a|s)(\log \pi_\theta(a|s) - \log \pi_{\theta^{old}|z_i}(a|s)) da \\
&= \frac{1}{N} \sum_{i=0}^{N-1} \int \pi_\theta(a|s) \nabla_\theta \log \pi_\theta(a|s)(\log \pi_\theta(a|s) - \log \pi_{\theta^{old}|z_i}(a|s)) da \\
&= \frac{1}{N} \sum_{i=0}^{N-1} \mathbb{E}_{a \sim \pi_{\theta|z}}[\nabla_\theta \log \pi_\theta(a|s)(\log \pi_\theta(a|s) - \log \pi_{\theta^{old}|z_i}(a|s))] \\
&= \frac{1}{N} \sum_{i=0}^{N-1} \mathbb{E}_{a \sim \pi_{\theta|z}}[\frac{1}{2} \nabla_\theta (\log \pi_\theta(a|s) - \log \pi_{\theta^{old}|z_i}(a|s))^2].
\end{aligned}
\tag{32}
$$

Empirically, we found little difference between fast dropout back-propagation and normal back-propagation when (32) is combined into (12).

## D  VARITAIONAL DROPOUT AND LOCAL REPARAMETRIZATION

As introduced in Section 2.1, multiplicative dropout at neuron activation of $k$th layer can also be viewed as a multiplicative noise for the input at $k + 1$th layer:

$$\mathbf{h}^{k+1} = \sigma(\mathbf{W}^{(k+1)T}\mathbf{D}_z^k\mathbf{h}^k + \mathbf{b}^{(k+1)}). \tag{33}$$

After a simple rearrangement $\tilde{\mathbf{W}}^{(k+1)T} = \mathbf{W}^{(k+1)T}\mathbf{D}_z^k$, we have:

$$\mathbf{h}^{k+1} = \sigma(\tilde{\mathbf{W}}^{(k+1)T}\mathbf{h}^k + \mathbf{b}^{(k+1)}). \tag{34}$$

That is, the stochastic neuron activation in networks with *Gaussian multiplicatice dropout* can also be regarded as stochastic neuron activation in Noisy Networks with correlated Gaussian parameter noise, whose means equal the corresponding ones in networks with dropout.

## E  HYPERPARAMETERS

For most of the hyperparamters, we follow the setting in original PPO paper. Though, to emphasis the scalability of our method, we use 2 parallel enviroments and only 1 minibatch in each epoch.

Table 1: Hyperparamters for PPO

| Hyperparameters | Value |
| --- | --- |
| Horizon | 2048 |
| Adam stepsize | $3 \times 10^{-4}$ |
| Num. epochs | 10 |
| Num. minibatch | 1 |
| Discount ($\gamma$) | 0.99 |
| GAE $\lambda$ | 0.95 |
| PPO clip | 0.2 |
| Num. layers | 2 |
| Num. hidden units | 64 |

## F  EXPERIMENT RESULTS IN STANDARD ENVIRONMENTS

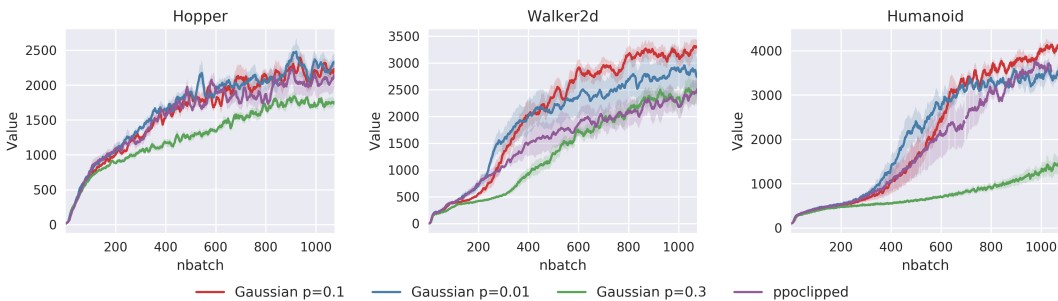

Figure 6: NADPEx in standard envs where *Gaussian dropout* is used

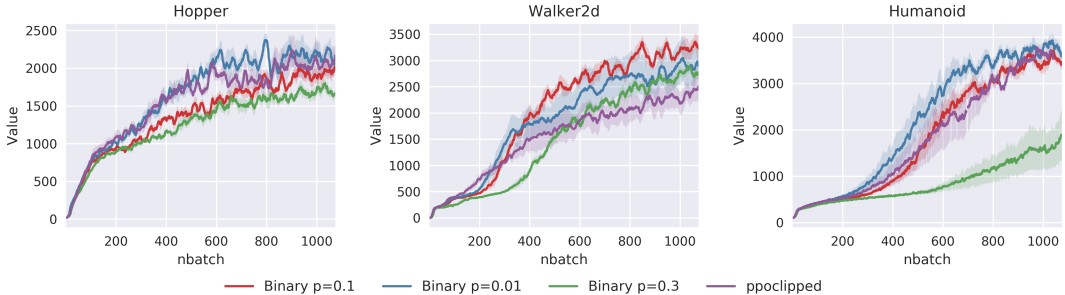

Figure 7: NADPEx in standard envs where *binary dropout* is used

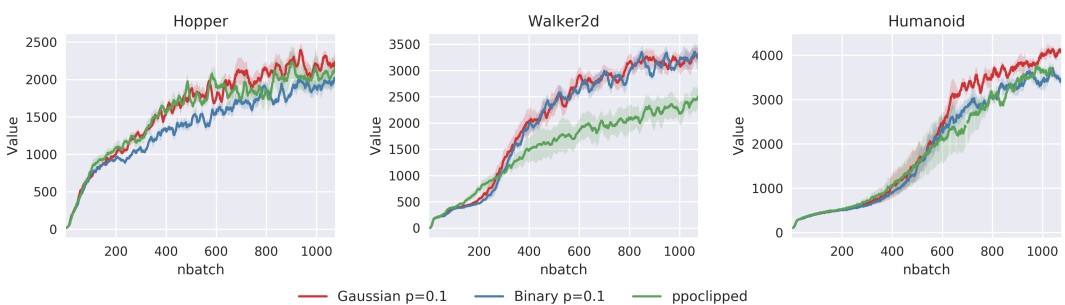

Figure 8: NADPEx in standard envs, camparing the best of *Gaussian dropout* and *binary dropout*

## G ENVIRONMENTS WITH SPARSE REWARDS

We use the same sparse reward environments from rllab Duan et al. (2016), modified by Houthooft et al. (2016):

- *SparseHalfCheetah* ($\mathcal{S} \subset \mathbb{R}^{17}, \mathcal{A} \subset \mathbb{R}^6$), which only yields a reward if the agent crosses a distance threshold,
- *SparseMountainCar* ($\mathcal{S} \subset \mathbb{R}^2, \mathcal{A} \subset \mathbb{R}$), which only yields a reward if the agent drives up the hill,
- *SparseDoublePendulum* ($\mathcal{S} \subset \mathbb{R}^6, \mathcal{A} \subset \mathbb{R}$), which only yields a reward if the agent reaches the upright position.

