# OpenReview forum: "NADPEx: An on-policy temporally consistent exploration method for deep reinforcement learning"
_ICLR.cc/2019/Conference_

### Official Review · AnonReviewer1 · 2018-11-02
**A nice paper on temporally consistent exploration**

**Rating:** 7
**Confidence:** 3

**Review:**

This paper proposed to use dropout to randomly choose only a subset of neural network as a potential way to perform exploration. The dropout happens at the beginning of each episode, and thus leads to a temporally consistent exploration. The paper shows that with small amount of Gaussian multiplicative dropout, the algorithm can achieve the state-of-the-art results on benchmark environments. And it can significantly outperform vanilla PPO for environments with sparse rewards.

The paper is clearly written. The introduced technique is interesting. I wonder except for the difference of memory consumption, how different it is compared to parameter space exploration. I feel that it is a straightforward extension/generalization of the parameter space exploration. But the stochastic alignment and policy space constraint seem novel and important.

The motivation of this paper is mostly about learning with sparse reward. I am curious whether the paper has other good side effects. For example, will the dropout cause the policy to be more robust? Furthermore, If I deploy the learning algorithm on a physical robot, will the temporally consistent exploration cause less wear and tear to the actuators when the robot explores. In addition, I would like to see some discussions whether this technique could be applied to off-policy learning as well.

Overall, I like this paper. It is well written. The method seems technically sound and achieves good results. For this reason, I would recommend accepting this paper.

---

> ### Author Response · Authors · 2018-11-10
> **Response to review**
>
> Glad to know that you like our paper!
>
> 1) Difference from parameter noise except for memory consumption:
> As stated in Section 3.3, we believe NADPEx is a generalization of parameter noise, with not only flexible memory consumption but also lower variance in gradients. This theory is examined in Section 4.2, where NADPEx shows faster convergence and lower variance in performance with different random seeds.
> Besides, comparing with [1], our work provides a theoretical modeling for the idea "a hierarchy of stochasticity for exploration". We model the NADPEx policy as a joint distribution of dropout random variables and actions, such that it could be combined seamlessly with existing on-policy policy gradient methods. One example is the policy space constraint stated in Section 3.2. We also provide another distribution i.e. Bernoulli distribution for stochasticity at high level, for which we derive gradient alignment and policy space constraint, as well as empirical results.
> As a minor point, in [1], the stochasticity at the high level i.e. the variance of parameter noise, is adjusted in a heuristic manner. NADPEx, in contrast, aligns the stochasticity throughout the hierarchy with end-to-end gradient update.
>
> 2) Other good side effects:
> The robustness of the NADPEx policy is orthogonal to our current work, but will be an interesting direction for the future. Currently we only have some preliminary results. For example, it is more robust to adversarial neural attacks. In the future we will investigate how robust NADPEx policies could be when the environment is perturbed, e.g. agents are dragged slightly by humans as in [2, 3].
> That temporally consistent exploration is fairly important for physical robots is one of our motivations for this whole project. In the next step we will look for simulator environments with more authentic actuators to see how NADPEx could help solve that. Our ultimate goal is to find a safer and more efficient way for on-policy exploration on physical robots.
> We believe the application of NADPEx to off-policy exploration is straightforward. However, as stated in Section 1, off-policy methods benefit from stronger flexibility for experience sampler. This makes the gradient alignment and policy space constraint not as important as in the on-policy methods. As off-policy methods have the potential to be much more data-efficient, we will compare in the future how NADPEx performs comparing with auto-correlated noise in [4] and separate sampler in [5].
>
> [1] Plappert et al., "Parameter Space Noise for Exploration", ICLR 2018.
> [2] Tassa et al., "Synthesis and stabilization of complex behaviors through online trajectory optimization", IROS 2012.
> [3] Clavera et al., "Learning to Adapt: Meta-Learning for Model-based Control", arXiv 2018.
> [4] Lillicrap et al., "Continuous control with deep reinforcement learning", ICLR 2016.
> [5] Xu et al., "Learning to explore via meta-policy gradient", ICML 2018.

---

### Official Review · AnonReviewer3 · 2018-11-05
**An interesting paper with unjustified approximations**

**Rating:** 6
**Confidence:** 3

**Review:**

The authors propose a new on-policy exploration strategy by using a policy with a hierarchy of stochasticity. The authors use a two-level hierarchical distribution as a policy, where the global variable is used for dropout. This work is interesting since the authors use dropout for policy learning and exploration.  The authors show that parameter noise exploration is a particular case of the proposed policy. The main concern is the gap between the problem formulation and the actual optimization problem in Eq 12. I am very happy to give a higher rating if the authors address the following points.

Detailed Comments
(1) The authors give the derivation for Eq 10. However, it is not obvious that how to move from line 3 to line 4 at Eq 15.
Minor:  Since the action is denoted by "a",  it will be more clear if the authors use another symbol to denote the parameter of q(z) instead of "\alpha" at Eq 10 and 15.

(2) Due to the use of the likelihood ratio trick, the authors use the mean policy as an approximation at Eq 12. Does such approximation guarantee the policy improvement? Any justification?

(3) Instead of using the mean policy approximation in Eq 12, the authors should consider existing Monte Carlo techniques to reduce the variance of the gradient estimation. For example, [1] could be used to reduce the variance of gradient w.r.t. \phi. Note that the gradient is biased if the mean policy approximation is used.

(4) Are \theta and \phi jointly and simultaneously optimized at Eq 12?  The authors should clarify this point.

(5) Due to the mean policy approximation, does the mean policy depend on \phi? The authors should clearly explain how to update \phi when optimizing Eq 12.

(6) If the authors jointly and simultaneously optimize \theta and \phi, why a regularization term about q_{\phi}(z)  is missing in Eq 12 while a regularization term about \pi_{\theta|z} does appear in Eq 12?

(7) The authors give the derivations about \theta such as the gradient and the regularization term about \theta (see, Eq 18-19). However, the derivations about \phi are missing.  For example, how to compute the gradient w.r.t. \phi? Since the mean policy is used, it is not apparent that how to compute the gradient w.r.t. \phi.
Minor, 1/2 is missing in the last line of Eq 19.

Reference:
[1] AUEB, Michalis Titsias RC, and Miguel Lázaro-Gredilla. "Local expectation gradients for black box variational inference." In Advances in neural information processing systems, pp. 2638-2646. 2015.

---

> ### Author Response · Authors · 2018-11-10
> **Response to review**
>
> Thank your very much for your review. We have updated the manuscript with more details in the derivation of the first order approximation of KL divergence.
>
> 1) Elaborated derivation of Eq. 10
> Q1: We have added one more line to explain the derivation. Basically a baseline is subtracted, and GAE is introduced.
>
> 2) Gradient update on \phi from KL divergence
> The gradients w.r.t. \phi from the KL divergence is stopped for variance reduction with acceptable bias, which we prove with MuProp [1]. Details could be found in Appendix C.
> Q3: Rather than [2], we employ MuProp to reduce variance in our development of NADPEx. Thank your for your suggestion.
> Q4: Yes \theta and \phi are jointly and simultaneously optimized at Eq. 12, though the gradients w.r.t. \phi from the KL divergence is stopped.
> Q7: Due to the stop-gradient manipulation in the KL divergence, gradients w.r.t. \phi remains the same as in stated in last subsection.
>
> 3) Mean policy in the KL divergence
> What motivates the mean policy is not variance reduction, but the idea that dropout policy had better to be close to each other. As intuitively \phi is controlling the distance between dropout policies, it would further remedy the little bias mentioned above. However, the computation complexity for "close to each other" would be O(N^2), with N being the number of dropout policies in this batch. We employ mean policy to make it linear. And it could be regarded as an integration on a Gaussian approximation of the Monte Carlo estimate according to [3]. Details could be found in Appendix C.
> Q2: No the mean policy is not used due to the likelihood ratio trick. And the approximation of using mean policy is discussed in [3], with a sound deduction.
> Q3: Mean policy is not motivated by variance reduction, which is addressed as introduced above. Thank you for your suggestion.
> Q5: In the updated version, we have explicitly pointed out that the gradients w.r.t. \phi from KL divergence is stopped. Thanks for this suggestion.
>
> Hope our response addresses your concerns!
>
> [1] Gu et al., "MuProp: Unbiased Backpropagation for Stochastic Neural Networks", ICLR 2016.
> [2] Titsias et al., "Local Expectation Gradients for Black Box Variational Inference", NIPS 2015.
> [3] Wang et al., "Fast dropout training", ICML 2013.

---

> > ### Comment · AnonReviewer3 · 2018-11-14
> > **Follow-up question (1)**
> >
> > "Q4: Yes \theta and \phi are jointly and simultaneously optimized at Eq. 12, though the gradients w.r.t. \phi from the KL divergence is stopped.
> > Q7: Due to the stop-gradient manipulation in the KL divergence, gradients w.r.t. \phi remains the same as in stated in last subsection."
> >
> > My guess is that due to the stop-gradient manipulation,  \phi remains the same when optimizing Eq 12. In other words, \phi is not updated. Is it correct? Can the authors comment on this?

---

> > > ### Comment · AnonReviewer3 · 2018-11-14
> > > **Follow-up  question (2)**
> > >
> > > On page 14 of the revised version, "To further reduce the variance, the first term is sometimes omitted with acceptable biased introduced (Raiko et al., 2015). "  Although the authors attempt to use the MuProp as justification, the first term is still ignored. Is it correct?
> > >
> > > Let's look at this term. (It is the last term of the last line at Eq 17)  as shown below.
> > > \nabla_{\phi} \int q_{\phi}(z) \int \pi_{\theta|z}(a|s) \log \frac{ q_{\phi}(z)  \pi_{\theta|z}(a|s) } {  q_{\phi^{old}}(z)  \pi_{\theta^{old}|z}(a|s)  } da dz (*)
> > >
> > > The authors argue that this term (*) should be ignored due to the high variance. Is it correct?
> > >
> > > Note that (*) can be decomposed into two terms as shown below:
> > > \nabla_{\phi} \int q_{\phi}(z)   \pi_{\theta|z}(a|s) \log \frac{ q_{\phi}(z)    } {  q_{\phi^{old}}(z)   }  dz                      (**)
> > > + \nabla_{\phi} \int q_{\phi}(z) \int \pi_{\theta|z}(a|s) \log \frac{   \pi_{\theta|z}(a|s)    } {  \pi_{\theta^{old}|z}(a|s)   } da dz     (***)
> > >
> > > (**) can be computed without using any samples since q_{\phi}(z)  is either a factorized Gaussian or factorized  Bernoulli distribution. In other words, the authors cannot ignore (**). Note that (**) is the KL term between q_{\phi}(z) and q_{\phi^{old}}(z).
> > >
> > > If the authors' reasoning is correct, including (**) into Eq 12 should also work.  Can the authors comment on this? Why (**) should be ignored? The authors should clarify this point.

---

> > > > ### Author Response · Authors · 2018-11-16
> > > > **Response to follow-up(2)**
> > > >
> > > > Q1. Even under the justification of MuProp, the estimate we provided is biased?
> > > > The idea of MuProp is to find a Taylor expansion of f(z), which is expected to be close to f(z). However, given the strong non-linearity in neural networks, we admit that there could be some bias introduced in this approximation. But this bias is believed to be acceptable in the literature of both probabilistic model and reinforcement learning. For example, as cited in Appendix C, [1] is a special case of MuProp [2] which reduces the variance with small bias. In the experiments of both [1] and [2], it is shown that this straight through gradient estimator outperforms unbiased estimators when there is only one layer of random variables, which is exactly the case for NADPEx. Another example is the Generalized Advantage Estimation (GAE) [3] in the reinforcement learning literature, which reduces the variance of high dimensional policy at a cost of bias. In Eq. 10 we use it in the second term, which basically reduces variance from the same cause as in Eq. 12: the high dimensionality of q_{\phi}.
> > > > We understand your concern that this bias may influence the effect of regularization to some extent. This concern drove us to further enforce the idea to make dropout policies to be close to each other. More details are introduced in Appendix C.
> > > >
> > > > Q2. Why in our development of the approximation do we calculate the KL divergence of q_{\phi} with Monte Carlo method given that it can be calculated analytically for both Gaussian and Bernoulli dropout?
> > > > Thanks for pointing it out, this is a brilliant discovery. We really appreciate your prudence in this review. The usage of Monte Carlo Estimate for KL divergence is inherited from PPO [4], which is introduced in Section 2.1 and Appendix A of our paper. Though not explicitly pointed out in their paper, in our empirical study we found that regularizing with an analytical KL tends to hinder the convergence. (Note that in continuous tasks actions are always modeled with diagonal Gaussian whose KL divergence also has analytical form.) We reckon that a Monte Carlo estimate of KL could soften this constraint, sharing the same motivation with the "trust region" in TRPO. (Without a trust region, this KL constraint may be too hard for an agent to acquire learning progress efficiently). Alternative to this Monte Carlo relaxation, there is [5] in the literature to combine the advantage of trust region relaxation and gradient co-optimization of KL divergence. Basically, the KL regularizer with trust region is implemented as an adaptive cutting mechanism. Gradients from surrogate loss will only be regularized with gradients from KL when the KL is larger than the trust region \delta. We provide another proof from this perspective that KL of q_{\phi} almost never violates this constraint of trust region in our updated version. That explains why gradients from KL(q_{\phi}||q_{\phi^{old}}) could be stopped in NADPEx even though it has analytical form.
> > > >
> > > > In sum, we feel grateful for your time and effort in helping us improve this paper. Our discussion has triggered us to deep-dive into the relaxation of KL divergence. We find it would be pretty interesting to investigate the relaxing effect of trust region and Monte Carlo in the future. In the same time, we want to emphasize that the possibility to conduct this theoretical analysis might be regarded as one of our work's contribution, as we provide a concrete form for NADPEx policies. NADPEx PPO is only one example. In the future, NADPEx could be combined with new on-policy policy gradient methods to help agent explore consistently and learn stably.
> > > >
> > > > [1] Raiko et al., "Techniques for learning stochastic feedforward neural networks", ICLR 2015.
> > > > [2] Gu et al., "MuProp: Unbiased Backpropagation for Stochastic Neural Networks", ICLR 2016.
> > > > [3] Schulman et al., "High dimensional continuous control with generalized advantage estimation", ICLR 2016.
> > > > [4] Schulman et al., "Proximal policy optimization algorithms", arXiv 2017.
> > > > [5] Wang et al., "Sample Efficient Actor-Critic with Experience Replay", ICLR 2017.

---

> > > ### Author Response · Authors · 2018-11-16
> > > **Response to follow-up(1)**
> > >
> > > As explained in our last response, \phi is still updated with gradients from surrogate loss i.e. Eq. 10 and Eq. 11. Note that if there was stop gradient operation, there would be two streams of gradients in NADPEx when a KL regularizer is added: one from surrogate loss, another one from KL divergence. We only stop gradients w.r.t. \phi from the KL divergence. "\phi is not updated" means gradients from surrogate loss are also stopped. Actually in our paper, it is referred to as bootstrap, named with BootstrapDQN [1], for which we provided a comparison with NADPEx in Section 4.3.
> > >
> > > [1] Osband et al., "Deep exploration via bootstrapped dqn", NIPS 2016.

---

### Official Review · AnonReviewer2 · 2018-11-05
**A novel on-policy exploration based on a distribution of plausible subnetworks and dropout strategy to achieve achieve on-policy temporally consistent exploration.**

**Rating:** 8
**Confidence:** 3

**Review:**

The authors introduce a  novel  on-policy  temporally  consistent  exploration  strategy, named Neural  AdaptiveDropout Policy Exploration (NADPEx), for deep reinforcement learning agents. The main idea is to sample from a distribution of plausible subnetworks modeling the temporally consistent exploration. For this, the authors use the ideas of the standard dropout for deep networks. Using the proposed  dropout transformation that is differentiable, the authors show that the KL regularizers on policy-space play an important role in stabilizing its learning. The experimental validation is performed on continuous control learning tasks, showing the benefits of the proposed.

This paper is very well written, although very dense and not easy to follows, as many methods are referenced and assume that the reviewer is highly familiar with the related works. This poses a challenge in evaluating this paper. Nevertheless, this paper clearly explores and offers a novel approach for more efficient on-policy exploration which allows for more stable learning compared to traditional approaches.

Even though the authors answer positively to each of their four questions in the experiments section,  it would like that the authors provide more intuition why these improvements occur and also outline the limitations of their approach.

---

> ### Author Response · Authors · 2018-11-10
> **Response to review**
>
> Thank you very much for your strong recommendation!
>
> 1) Intuition about the improvement
> Though not explained in Section 4. The intuition for NADPEx is given in Section 3. Interpretation for as efficient or even faster exploration in dense environment (4.1) is that NADPEx could encourage more diverse exploration, while absorb experience from it in a relatively efficient way. For sparse environments (4.2), where temporally consistent exploration is crucial for learning signal acquisition, NADPEx outperforms vanilla PPO. It could also beat parameter noise if difficulty is increased, because intuitively low variance in gradients is a boon for faster learning. Improvement in 4.3 and 4.4 are basically from the theoretical grounding of NADPEx, which we believe is one of our contributions. Specifically, improvement in 4.3 is from high level stochasticity's adaptation to the low level; while that in 4.4 could be interpreted with the idea of trust region, that policy should be updated to somewhere near the sampling policy in the policy space, such that collected experience are usable (on-policy).  In NADPEx, trust region also contains the meaning that dropout policies are close to each other for more efficient exploration.
>
> 2) Limitation of NADPEx
> One of the limitation we see from NADPEx is that dropout policies are not directly interpretable from their network structures, while interpretability and composibility are prerequisites for reusing them in more complicated tasks. Luckily, modeled as latent random variables, an information term could be added to the objective as in [1, 2]. This is also a direction for future research work.
>
> [1] Florensa et al., "Stochastic neural networks for hierarchical reinforcement learning", ICLR 2017.
> [2] Hausman et al., "Learning an Embedding Space for Transferable Robot Skills", ICLR 2018.

---

### Public Comment · (anonymous) · 2018-10-30
**Regarding Sparsity**

I enjoyed your paper, and I have two questions.

1. In page 7, you 'gradually increase the difficulty' of three sparse-reward environments.
Could you explain in detail about this sentence?
I want to know the period of increase and the corresponding threshold values for SparseDoublePendulum, HalfCheetah. and MountainCar.

2. Unlike 4.1 and 4.4 in which 'large initial dropout rate may induce large variance',
why is it that large initial dropout rate was helpful for highest performance in 4.2?

---

> ### Author Response · Authors · 2018-10-30
> **Details about sparse environments**
>
> Thank you for your questions!
>
> 1. 'gradually increase the difficulty'
> By 'gradually increase' we mean we ran experiments repeatedly with increasing difficulty, while the difficulty remains fixed in each experiment.  And the motivation is to amplify the difference between NADPEx and parameter noise. In the three listed files under directory rllab/envs/sparse_envs, the difficulty is denoted with 'PROP', whose default value is 1. We have tried incrementing it by 0.1 in each repetition, and the final value is 2.
>
> 2. large initial dropout rate
> The difference between 4.1/4.4 and 4.2 is the density of reward.
> When the reward is dense (4.1, 4.4), temporally consistent exploration is not the crucial obstacle for the agent to acquire learning signals. (Though our experiments still reveal that a little temporally consistent dropout could help to reach a better optimum). In this circumstance, what matters more is the speed of convergence for stochastic policy optimization. A large initial dropout rate promotes high variance in gradients, a condition all stochastic neural networks try to avoid [1, 2]. Therefore, it is highly possible that NADPEx agents over-explore when the initial dropout rate is high, not making full use of their experience.
> On the other hand, when the reward is sparse, the role played by temporally consistent exploration becomes significant. With larger initial dropout rate, NADPEx policies possess higher stochasticity and thus exhibit more diverse behaviors. In sparse environment, agents with diverse behaviors are more likely to discover useful learning signals, requiring less data to converge to an optimum. While those with small initial dropout rates are prone to under-exploration, taking longer time to collect the useful information.
>
> [1] Gu et al., "Mu-prop: Unbiased back-propagation for stochastic neural networks", ICLR 2016.
> [2] Gu et al., "Q-prop: Sample efficient policy gradient with an off-policy critic", ICLR 2017.

---

### Author Response · Authors · 2018-11-10
**Manuscript updated**

We thank all reviewers for your comments and recommendation! We have updated the manuscript in the following sections, taking your feedbacks into account:
1) In Section 3.2 POLICY SPACE CONSTRAINT, we make a clarification on  the omission of gradient w.r.t. \phi from KL divergence and the replacement with mean policy. Basically they serve for two separate concerns, variance reduction and a remedy from \theta to this omission. In the previous version we mingled them together to make it simple and intuitive, sorry for the confusion it caused.
2) In Appendix C, we give a detailed derivation of this approximation for both binary dropout and Gaussian dropout,  providing a proof that it makes the training robust and stable with little and acceptable bias.
3) In Section 4.2, we elaborate the expression "gradually increase the difficulty".

---

> ### Author Response · Authors · 2018-11-16
> **Manuscript updated (2)**
>
> We have updated Appendix C to provide an alternative approximation deduction in response to AnonReviewer3's follow-up.

---

### Meta-Review · Area_Chair1 · 2018-12-15
**meta-learning**

**Confidence:** 2
**Recommendation:** Accept (Poster)

**Metareview:**

The authors have proposed a new method for exploration that is related to parameter noise, but instead uses Gaussian dropout across entire episodes, thus allowing for temporally consistent exploration. The method is evaluated in sparsely rewarded continuous control domains such as half-cheetah and humanoid, and compared against PPO and other variants. The method is novel and does seem to work stably across the tested tasks, and simple exploration methods are important for the RL field. However, the paper is poorly and confusingly written and really really needs to be thoroughly edited before the camera ready deadline. There are many approaches which are referred to without any summary or description, which makes it difficult to read the paper. The three reviewers all had low confidence in their understanding of the paper, which makes this a very borderline submission even though the reviewers gave relatively high scores.